# A miniature CRISPR–Cas10 enzyme confers immunity by inhibitory signalling

Erin E. Doherty[1,2,3,11], Benjamin A. Adler[1,2,3,11], Peter H. Yoon[1,2], Kendall Hsieh[2], Kenneth Loi[2], Emily G. Armbruster[4], Arushi Lahiri[2], Cydni S. Bolling[1], Xander E. Wilcox[5], Amogha Akkati[6], Anthony T. Iavarone[3], Joe Pogliano[4] & Jennifer A. Doudna[1,2,3,7,8,9,10]✉

Microbial and viral co-evolution has created immunity mechanisms involving oligonucleotide signalling that share mechanistic features with human antiviral systems[1]. In these pathways, including cyclic oligonucleotide-based antiphage signalling systems (CBASSs) and type III CRISPR systems in bacteria and cyclic GMP–AMP synthase–stimulator of interferon genes (cGAS–STING) in humans, oligonucleotide synthesis occurs upon detection of virus or foreign genetic material in the cell, triggering the antiviral response[2–4]. Here, in an unexpected inversion of this process, we show that the CRISPR-related enzyme mCpol synthesizes cyclic oligonucleotides constitutively as part of an active mechanism that represses a toxic effector. Cell-based experiments demonstrated that the absence or loss of mCpol-produced cyclic oligonucleotides triggers cell death, preventing the spread of viruses that attempt immune evasion by depleting host cyclic nucleotides. Structural and mechanistic investigation revealed mCpol to be a di-adenylate cyclase whose product, c-di-AMP, prevents toxic oligomerization of the effector protein 2TMβ. Analysis of cells by fluorescence microscopy showed that lack of mCpol allows 2TMβ-mediated cell death due to inner membrane collapse. These findings unveil a powerful defence strategy against virus-mediated immune suppression, expanding our understanding of the role of oligonucleotides in immunity.

Oligonucleotide-based immune signalling is conserved across the tree of life[1]. In this strategy, a signalling enzyme senses viral infection and synthesizes an oligonucleotide product to activate an immune response. Studies of type III CRISPR–Cas systems first demonstrated that viral infection induces synthesis of cyclic oligonucleotides in prokaryotes to activate effectors that restrict phage[5,6]. Subsequently, the discovery of prokaryotic CBASSs revealed a wide diversity of signalling nucleotides and effector proteins that expand this defence strategy[3,7,8]. These advances paralleled the previous identification of the eukaryotic cGAS–STING pathway[2,9], underscoring the deep conservation of oligonucleotide signalling as a mechanism of antiviral defence[3,4,7,10–14].

Distinct pathways for oligonucleotide-based immune signalling in bacteria have been tentatively linked by an uncharacterized gene encoding minimal CRISPR polymerase (mCpol)[15] that often co-occurs with CBASS genes and shares sequence homology with *cas10*, the signature gene of type III CRISPR systems[16] (Fig. 1a). Bioinformatic studies have noted these associations, raising the hypothesis that mCpol could represent a polymerase adapted for defence, but its function has not been experimentally demonstrated[15]. These associations suggested that mCpol might contribute to bacterial immunity by an unknown mechanism.

Here we show that mCpol and its associated 2TMβ effector protein, hereon referred to as Panoptes[17], confer antiphage immunity in a process that detects viral anti-immunity enzymes by inverting the role of signalling oligonucleotides. Instead of oligonucleotide-induced cell death by effector activation in response to viral infection, mCpol constitutively synthesizes 2′3′-cyclic diadenylate (c-di-AMP) that prevents toxic 2TMβ oligomerization. Disruption of c-di-AMP production in mCpol–2TMβ-containing cells, such as occurs with phage-encoded anti-CBASS (Acb) sponges[18–20] and phosphodiesterases[21], allows 2TMβ multimerization and cell death by inner membrane collapse. Panoptes systems[17] co-locate with CBASS immune systems in bacterial genomes to detect phage anti-immunity enzymes and trigger host cell destruction.

## mCpol contributes to antiphage defence

While investigating the genomic locus surrounding the Hachiman antiphage defence system in *Escherichia coli* ECOR31 (ref. 22), we observed an adjacent operon encoding mCpol and a 2TMβ protein: Panoptes (Fig. 1a). To test whether mCpol has an independent role in antiphage immunity, we expressed the Panoptes operon, containing

[1]Innovative Genomics Institute, University of California, Berkeley, Berkeley, CA, USA. [2]Department of Molecular and Cell Biology, University of California, Berkeley, Berkeley, CA, USA. [3]California Institute for Quantitative Biosciences (QB3), University of California, Berkeley, Berkeley, CA, USA. [4]School of Biological Sciences, University of California San Diego, La Jolla, CA, USA. [5]Department of Microbiology and Immunology, Cornell University, Ithaca, NY, USA. [6]Department of Plant and Microbial Biology, University of California, Berkeley, Berkeley, CA, USA. [7]Gladstone Institutes, University of California, San Francisco, San Francisco, CA, USA. [8]Howard Hughes Medical Institute, University of California, Berkeley, Berkeley, CA, USA. [9]Department of Chemistry, University of California, Berkeley, Berkeley, CA, USA. [10]MBIB Division, Lawrence Berkeley National Laboratory, Berkeley, CA, USA. [11]These authors contributed equally: Erin E. Doherty, Benjamin A. Adler. ✉e-mail: doudna@berkeley.edu

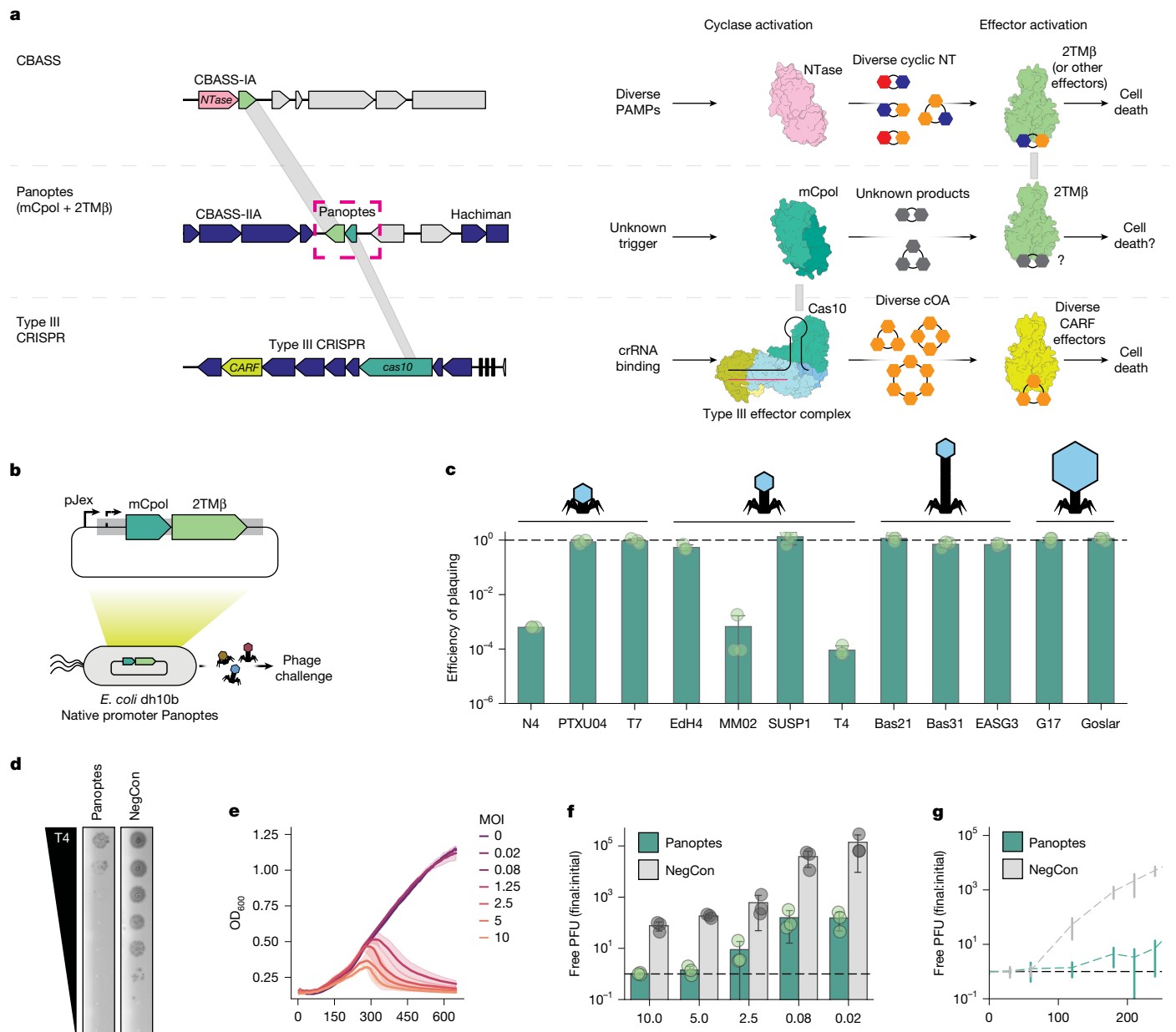

**Fig. 1 | Effects of mCpol and 2TMβ expression on *E. coli* susceptibility to bacteriophage infection. a**, Representative CBASS (top left; NZ_PTRT01000132), Panoptes (top middle; NZ_LYAF01000004)[15] and type III-A CRISPR–Cas loci (top bottom; NC_002976). Shared protein domains (grey) between CBASS and Panoptes (2TMβ and Cap15, 23% identity) and between Panoptes and type III CRISPR loci (mCpol and Cas10, 19% identity) are shown. CBASS, Panoptes and type III CRISPR signalling in response to pathogen-associated molecular patterns (PAMPs; right). NT, nucleotides. **b**, Phage challenge experiments. The Panoptes locus from ECOR31 is expressed on a low-copy plasmid under its native promoter. **c**, Change in efficiency of plaquing when phages were challenged against *E. coli* with the ECOR31-derived Panoptes locus, normalized to a vector control (black dashed line). Data shown represent the mean of three biological replicates ± standard deviation. Cartoons represent different bacteriophage morphologies (left to right: podovirus-like,

myovirus-like, siphovirus-like and jumbophage). **d**, Plaque assay in which Panoptes or a negative control (NegCon) is challenged with serial dilutions of phage T4. Representative of three biological replicates. The efficiency of plaquing is quantified in panel **c**. **e**, Cell density over time of *E. coli* with a plasmid expressing the Panoptes locus at various MOIs. Data shown represent the mean of three biological replicates ± standard deviation. **f**, T4 phage production for *E. coli* with a plasmid containing the Panoptes system or a vector control at increasing MOI. Data are presented as the mean of three biological replicates ± standard deviation. The black dashed line represents no additional production of phages. **g**, One-step growth curve of T4 infection Panoptes and a vector control over time (in minutes post-induction), representing one round of infection at approximately 0.01 MOI. Data are are the mean of three biological replicates ± standard deviation. The black dashed line represents no additional production of phages.

just the mCpol-encoding and 2TMβ-encoding genes, using its native promoter in *E. coli* and challenged the strain with 12 different diverse bacteriophages (Fig. 1b). Although most phage killed the cells in the presence of Panoptes, expression of Panoptes provided $10^3$–$10^4$-fold protection from phages T4, MM02 and N4 (Fig. 1c,d and Extended Data Fig. 2). To determine the mode of protection, we exposed the

Panoptes-expressing strain to phage T4 at various multiplicities of infection (MOIs). High MOIs (1.25 or more infections per cell) resulted in limited phage production but also significant cell death (Fig. 1e,f). At low MOI (0.08 or lower infections per cell), however, Panoptes limited T4 production in early rounds of infection to provide population-level antiphage protection (Fig. 1g). Together, these results suggest that

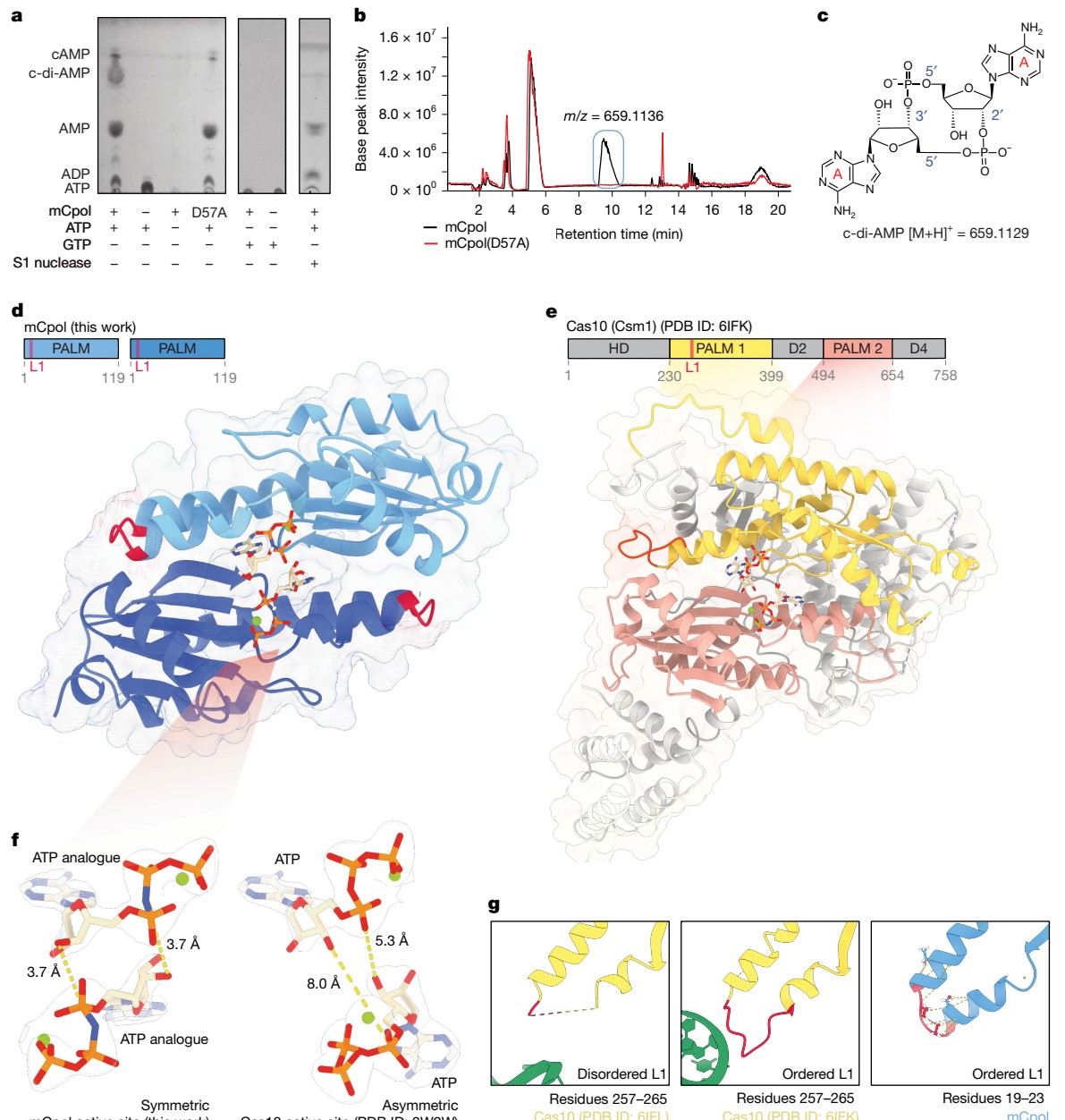

**Fig. 2 | Crystal structure of mCpol and biochemical detection of adenylate cyclase activity. a**, TLC analysis of product formation upon reaction with mCpol. Additional products are formed when mCpol is incubated with ATP. The primary product of the mCpol–ATP reaction is not formed by mCpol(D57A). No product is formed when mCpol is incubated with GTP. The primary product of the mCpol–ATP reaction is degraded by S1 nuclease. **b**, Differential HPLC–MS reveals a molecular weight of 659.1136 *m/z* for the D57A-dependent product, which is consistent with c-di-AMP. **c**, Structure and molecular weight of the primary mCpol product as determined by TLC, S1 digestion and HPLC: 2′3′-c-di-AMP. **d**, X-ray crystal structure of mCpol bound to ApNHpp (PDB ID: 9NWN). **e**, Domain map and structure of Cas10 (Csm1) highlighting PALM domains

conserved between mCpol and Cas10 in pink and yellow (PDB ID: 6IFK). The loop L1 region in the PALM 1 domain is highlighted in pink on the domain map and structure. **f**, Positioning of 2 ATP analogues (ApNHpp) in the mCpol active site (this work) and 2 ATP in the Csm1 active site (donor/acceptor; PDB ID: 3W2W) showing symmetric versus asymmetric positioning of the ATP molecules. Dashed lines indicate atomic distances. **g**, Loop L1 of Csm1 (residues 257–265) is disordered (dashed yellow line indicates residues not modelled in the structure) in the presence of non-cognate target RNA (PDB ID: 6IFL) and ordered in the presence of cognate target RNA (PDB ID: 6IFK). The region of mCpol corresponding to loop L1 (residues 19–23) consists of a highly ordered turn stabilized by intramolecular hydrogen bonds (dashed lines).

Panoptes confers phage defence through an abortive infection phenotype[23] consistent with its encoding of a Cap15 effector homologue (2TMβ), which multimerizes to collapse the inner membrane of the cell during phage infection[24].

Having established Panoptes as an antiphage system, we next investigated the basis of Panoptes signalling. An alignment of 163 mCpol proteins revealed a semi-conserved di-glycine di-aspartate active site residues consistent with known homology to Cas10 and related

nucleotidyl cyclases that catalyse ATP dicyclization[15,16,25] (Supplementary Fig. 2). To assess whether mCpol has the intrinsic ability to synthesize cyclic oligonucleotides, recombinant mCpol was incubated in a buffer containing adenosine or guanosine triphosphate (ATP or GTP). Analysis of these reactions by thin-layer chromatography (TLC) showed the presence of new species in the ATP-containing reactions but not those containing GTP (Fig. 2a and Supplementary Fig. 1). Mutation of the strictly conserved aspartate residue (D57A) in mCpol completely

eliminated production of one new species (Fig. 2a and Supplementary Fig. 2).

To determine the identity of the primary reaction product of mCpol, we used differential high-performance liquid chromatography–mass spectrometry (HPLC–MS) to show that the single species present with wild-type mCpol but absent with the D57A mutant has a molecular weight consistent with cyclic di-adenosine monophosphate (c-di-AMP; Fig. 2b,c and Extended Data Fig. 2). Treatment of mCpol–ATP reaction products with S1 nuclease, which specifically cleaves 5′–3′ phosphodiester linkages, depleted the reaction c-di-AMP (Fig. 2a). MS of the S1 nuclease digestion product revealed a molecular weight consistent with linear 5′-phosphoadenylyl adenosine (pApA), indicating the presence of one 5′–3′ phosphodiester linkage (Extended Data Fig. 2). To determine the product identity, HPLC–MS of the mCpol–ATP reaction was compared with cyclic dinucleotide standards to reveal that the major product of mCpol is 2′3′-c-di-AMP (Extended Data Fig. 2). These results reveal that mCpol synthesizes 2′3′-c-di-AMP from ATP, mirroring the selectivity of Cas10 for the adenine base[8]. Of note, however, mCpol makes a singular c-di-AMP product, whereas Cas10 makes an array of higher-order cyclic oligoadenylates ($cOA_n$; $n = 2–6$)[7].

## mCpol signalling inhibits 2TMβ effector

Previously studied antiviral second messenger synthases are tightly regulated sensors that become active during infection[26]. For example, cOA synthesis by Cas10 depends on cognate target RNA recognition[27]. However, we observed that c-di-AMP synthesis by mCpol proceeded in vitro without an additional stimulus. To determine how the regulation of mCpol may diverge from that of Cas10, we solved a high-resolution X-ray crystal structure of mCpol bound to a non-hydrolysable ATP analogue (ApNHpp; Fig. 2d and Extended Data Figs. 1 and 3). This structure showed mCpol to contain a Cas10-like PALM domain that dimerizes to form a pocket where two ATP analogues are bound[28] (Fig. 2d,e). A conserved serine, also seen in Cas10, is one of two residues (S30 and N34) that confer ATP-binding selectivity (Extended Data Fig. 3). However, the positioning of ATP substrates in the active site differs substantially between mCpol and Cas10. Previous structures of Cas10 bound to ATP revealed the presence of asymmetric donor and acceptor molecules positioned to allow for a stepwise formation of each linkage[27,29] (Fig. 2f). In the mCpol active site, however, symmetry between the two ATP analogues positions them for coordinated synthesis of both linkages to form c-di-AMP (Fig. 2f). This substrate orientation explains the lack of higher-order cOAs, which are synthesized by Cas10.

Previous studies have shown that loop L1 of the Cas10 PALM1 domain (*Streptococcus thermophilus* Csm1 residues 257–265) acts as a critical allosteric regulator of cyclase activity[30]. This regulatory loop bridges two α-helices, one of which borders the active site residues. Residues in Cas10 loop L1 are disordered in the presence of non-cognate target RNA, which does not activate Cas10, but become ordered upon binding of cognate target RNA[30] (Fig. 2g). In the structure of mCpol, however, a highly ordered turn that is stabilized by a network of hydrogen bonds replaces the region corresponding to loop L1 (residues 19–23) of Cas10. This structural difference suggests that mCpol adopts an active conformation without additional requirements such as RNA binding (Fig. 2g). The absence of homology to the regulatory region of Cas10 (ref. 25), and the stabilization of the loop L1-equivalent structure, are consistent with our observation that mCpol is a constitutively active cyclase.

We wondered how constitutive c-di-AMP synthesis enables an immune response specific to phage infection. Early in our investigation of mCpol-mediated phage defence, we were surprised to observe that increased expression of mCpol–2TMβ conferred reduced phage defence (Extended Data Fig. 4), as increased expression of the CBASS nucleotidyltransferase and effector has the opposite effect of providing more robust phage defence and background toxicity[24]. On the basis of our observations of constitutive mCpol activity and increased

mCpol expression leading to decreased phage defence and cell death, we hypothesized that the Panoptes system acts by mCpol-mediated negative regulation, akin to toxin–antitoxin (TA) systems[31].

To test for a toxin–antitoxin relationship between mCpol and 2TMβ, we separated the Panoptes operon into two plasmids, where orthogonal inducers controlled expression of mCpol and 2TMβ (Fig. 3a). In the absence of mCpol, we found that 2TMβ expression had a deleterious effect on cell viability and growth, suggesting its role as a toxin (Fig. 3c). mCpol expression suppressed the growth defects incurred by 2TMβ expression, suggesting that it has a role in neutralizing 2TMβ (Fig. 3b). No suppression of 2TMβ toxicity occurred upon co-expression of an mCpol catalytic mutant (Fig. 3d). These data indicate that Panoptes is a toxin–antitoxin system in which the catalytic activity of mCpol prevents 2TMβ toxicity.

## c-di-AMP blocks 2TMβ-induced cell lysis

The observation that active mCpol suppresses the toxicity of 2TMβ suggested that the c-di-AMP product of mCpol operates as the cognate antitoxin. AlphaFold 3 predicts 2TMβ to comprise N-terminal transmembrane helices and a C-terminal β-barrel domain in an architecture mirroring that of the CBASS effector protein Cap15 (Fig. 3e). Upon cyclic dinucleotide binding to its β-barrel domain, Cap15 multimerizes and induces cell death by disrupting inner-membrane integrity[24]. The strong homology between Cap15 and the predicted 2TMβ structure (Dali $Z$ score of 16, where more than 2 indicates significant homology), and the TMHMM prediction of transmembrane topology indicate that 2TMβ would similarly display membrane localization[32–34] (Supplementary Fig. 3). We noted that residues involved in cyclic oligonucleotide binding by Cap15 are conserved in 2TMβ (Supplementary Fig. 4), suggesting that c-di-AMP binding may directly control 2TMβ activity. To test the hypothesis that mCpol-generated c-di-AMP inhibits 2TMβ multimerization-mediated cell death, we examined the effect of c-di-AMP on purified 2TMβ. The β-barrel domain of 2TMβ (2TMβ(Δ2TM)) was overexpressed in the presence of mCpol and purified by size-exclusion chromatography with supplementation of 2′3′-c-di-AMP in all buffers (Fig. 3e and Extended Data Fig. 5a,b). By dialysing away 2′3′-c-di-AMP, we were able to probe the binding of 2TMβ(Δ2TM) to the signalling molecule. A thermal melting assay showed significant stabilization of 2TMβ(Δ2TM) upon addition of 2′3′-c-di-AMP, indicating that the 2′3′-c-di-AMP ligand is directly bound by the protein[24,35] (Extended Data Fig. 5c). Correspondingly, the apo 2TMβ(Δ2TM) protein demonstrates an oligomeric state by analytical size-exclusion chromatography that is consistent with the molecular weight observed by native PAGE (Fig. 3f and Extended Data Fig. 5d,e). Titrating 2′3′-c-di-AMP into the recombinant 2TMβ(Δ2TM) sample leads to an eventual loss of the multimeric species (Fig. 3f). These results illustrate that, rather than multimerizing in response to the cyclic dinucleotide like the CBASS Cap15 effector, 2TMβ multimerizes in the absence of 2′3′-c-di-AMP (Fig. 3g).

We next analysed the basis for toxicity of 2TMβ by assessing bacterial cell integrity under different conditions using fluorescence microscopy (Fig. 3h–k). When 2TMβ expression alone was induced in our co-expression strain (Fig. 3i), cells developed membrane lesions, most often at the cell poles. Consistent with our cell viability assays (Fig. 3b,c), membrane lesions did not develop when mCpol alone or both mCpol and 2TMβ were expressed (Fig. 3i) or in our negative control strain carrying an empty vector instead of the 2TMβ plasmid (Fig. 3j). Cells lacking the mCpol plasmid (2TMβ only) developed membrane lesions and frequently lysed (Fig. 3k and Extended Data Fig. 6a–c). Collectively, these results demonstrate that mCpol signalling prevents cell death by inhibiting 2TMβ multimerization-mediated inner membrane collapse. This inversion of typical regulatory logic deviates from all known immune signalling mechanisms in which virus-triggered nucleotide signals activate an immune effector.

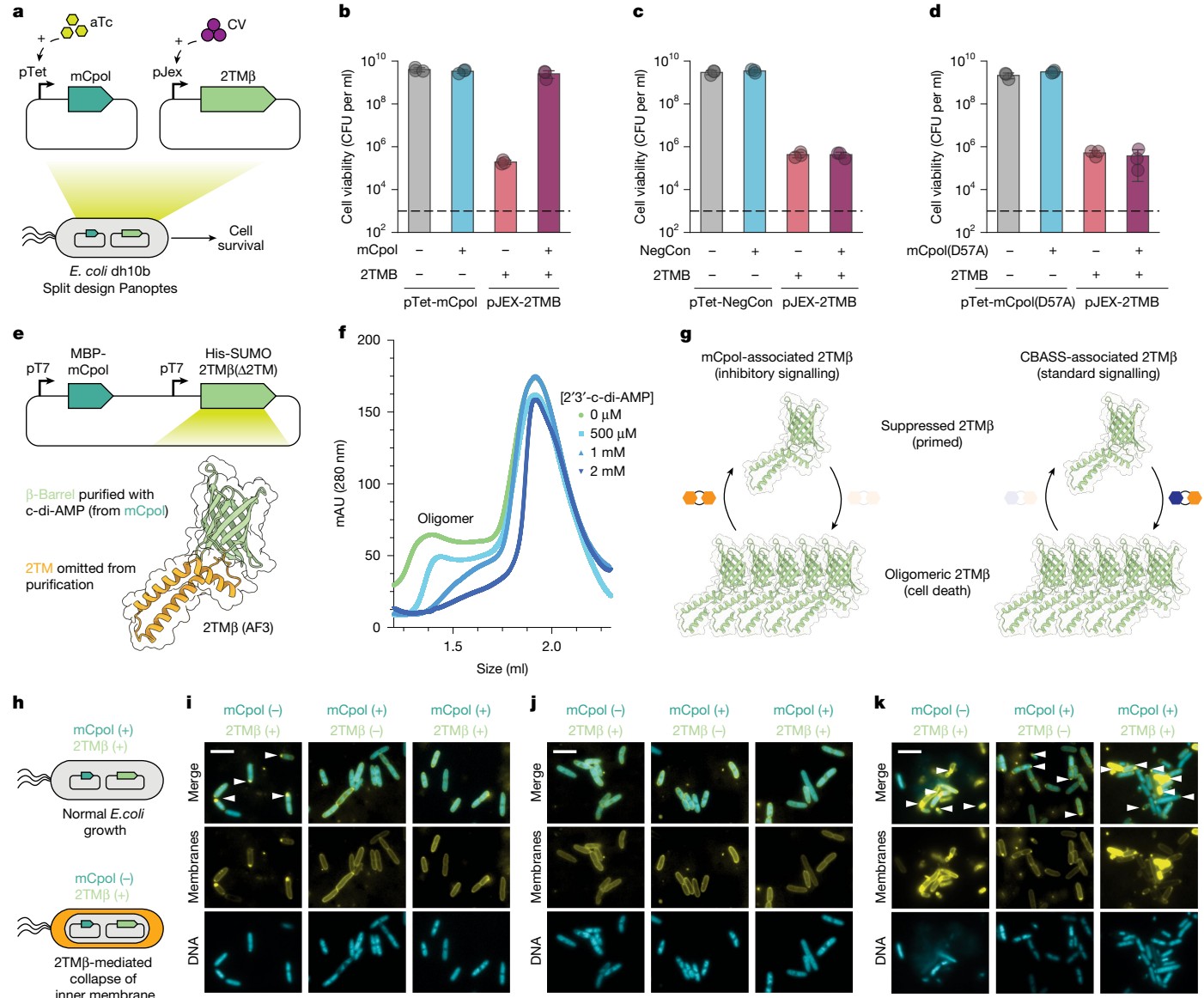

**Fig. 3 | Effects of mCpol-generated c-di-AMP on 2TMβ protein behaviour.**
**a**, Overview of the mCpol split design to investigate toxin–antitoxin dynamics in *E. coli*. **b**–**d**, Cell viability in colony-forming units (CFU) per ml when plasmid is present but no expression is induced (−) or expression has been induced (+) by conditions in Fig. 3a. Toxin and antitoxin components are expressed under a pJex promoter with a 0 nM or 50 nM crystal violet inducer and under a pTet promoter with a 0 nM (−) or 20 nM (+) anhydrotetracycline inducer (aTc). Experiments represent the mean of three biological replicates ± standard deviation. Dashed lines represent limit of detection. **e**, Approach for expression of 2TMβ(Δ2TM) in the presence of mCpol. The model was generated by AlphaFold 3 (AF3). **f**, Incubation of 4.75 mg ml⁻¹ (323 μM) apo 2TMβ(Δ2TM) with increasing concentrations of 2′3′-c-di-AMP as visualized by analytical size-exclusion chromatography. Concentrations of 2′3′-c-di-AMP ranged from 500 μM to 2 mM. The presence of a peak at less than 1.5 ml indicates a multimer that dissipates as the concentration of 2′3′-c-di-AMP is increased. **g**, Contrasting mechanisms of Panoptes-associated 2TMβ (left) and CBASS-associated 2TMβ (right). mCpol-associated 2TMβ shows oligomerization in the absence of the signalling molecule, whereas CBASS-associated 2TMβ signalling shows oligomerization in the presence of the signalling molecule[24]. **h**, Bacterial physiology of inner membrane integrity in the presence (top) or absence (bottom) of mCpol expression. **i**–**k**, Fluorescence microscopy images for leaky (−) and induced (+) expression of mCpol, 2TMβ or a vector control. DNA is stained with 4′,6-diamidino-2-phenylindole (teal) and all membranes are stained with MitoTracker Green (yellow). The arrowheads highlight polar membrane collapse or lack thereof. Scale bars, 5 μm. For microscopy experiments (**i**–**k**), mCpol and candidate toxin are expressed by induction with 0 nM or 200 nM aTc and 0 nM or 50 nM crystal violet (CV), respectively. All microscopy experiments (**i**–**k**) were performed in biological duplicate.

## Acb proteins activate 2TMβ toxicity

The inhibitory signalling logic encoded by Panoptes suggested that disruption of immune signalling could trigger 2TMβ-induced cell death. Of note, phages sensitive to Panoptes encode Acb proteins that interfere with immune signalling by hydrolysing or sequestering cyclic oligonucleotides (Fig. 1c). T4 and MM02 harbour the cyclic nucleotide degrader Acb1 (encoded by *T457B* and *MM02gp156*)[21] and cyclic nucleotide sponge Acb2 (encoded by *T4vs.4* and *MM02gp117*)[18].

N4 harbours the cyclic nucleotide sponge Acb4 (*N4gp48*)[20]. Furthermore, three T4-like phages that are naturally resistant to Panoptes encode loss-of-function mutations in *acb1* or *acb2* (refs. 35–38) (Extended Data Fig. 7).

To test whether Acb1_T4, Acb2_T4 and Acb4_N4 are necessary for Panoptes activation, we used CRISPR interference through antisense RNA targeting (CRISPRi-ART)[39]. CRISPRi-ART uses dCas13d RNA targeting to mediate gene expression knockdown in phage during infection (Fig. 4a and Extended Data Fig. 8a–c). CRISPRi-ART elimination of Acb

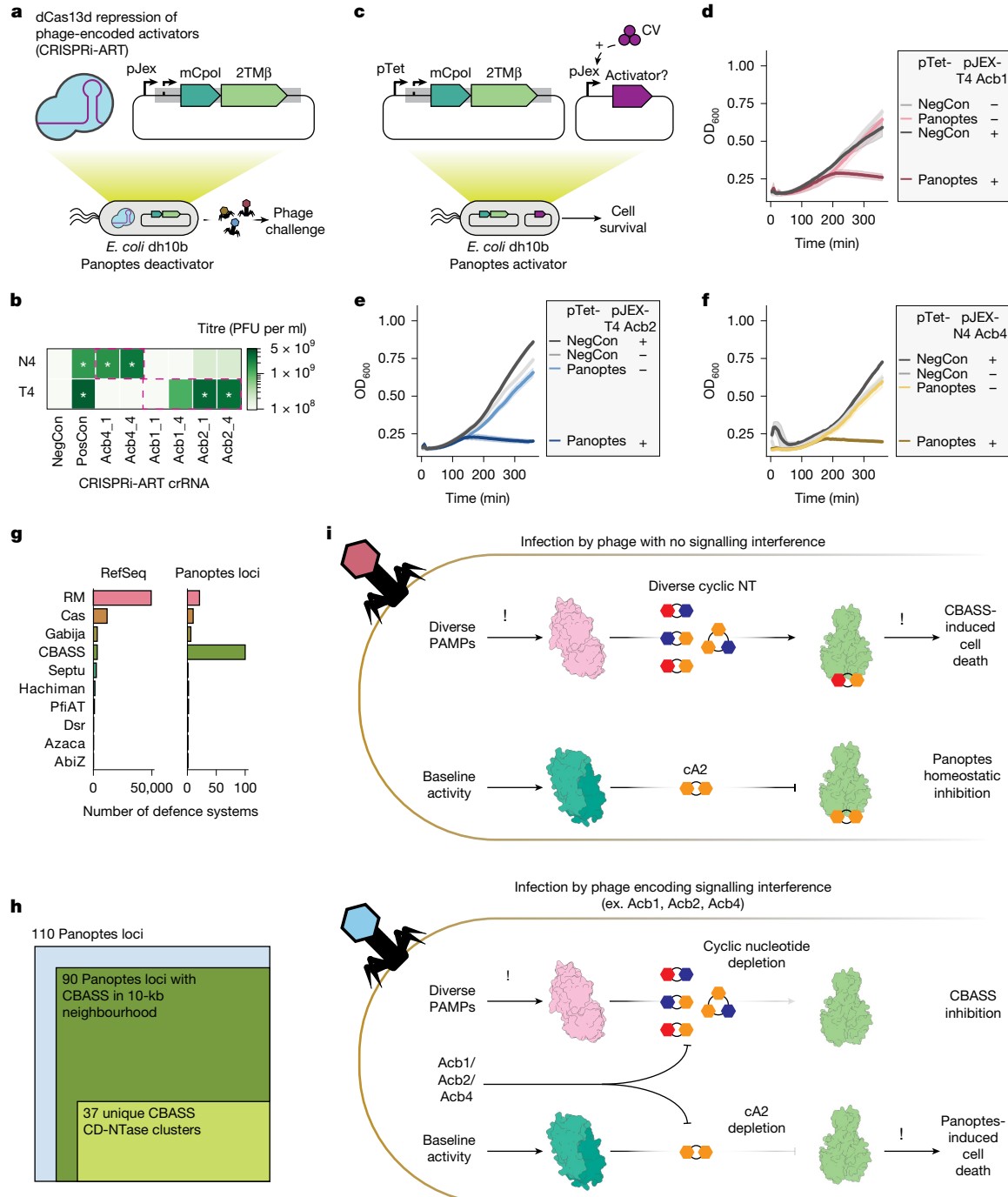

**Fig. 4 | Panoptes activation by phage-encoded Acb proteins. a**, CRISPRi-ART screening of phage-encoded Acb proteins by knockdown with dCas13d (blue protein with purple guide RNA) to determine phage infectivity. **b**, Change in plaque-forming units (PFU per ml) during knockdown of Acb proteins. CRISPRi-ART was expressed at 20 nM aTc. The $x$ axis indicates the protein being targeted (Acb4, Acb2 or Acb1) and the guide RNA ('gRNA1' or 'gRNA4'). Controls represented are a non-targeting crRNA in the presence (NegCon) or absence (PosCon) of Panoptes. Restoration of plaques to PosCon size is marked with white asterisks. CRISPRi-ART conditions targeting a phage-encoded Acb protein in the target phage are bounded in magenta dashed lines. Phage PFU per ml are reported as the mean of three biological replicates. crRNA, CRISPR RNA. **c**, Activator assays (**d**–**f**) with Panoptes under control of its native promoter. The plasmid under pTet control (Panoptes or vector control (NegCon); left), and the second plasmid, under the control of a pJEx promoter, whether induced (+)

at 125 nM crystal violet (CV) or not induced (−; right) are shown. Data shown represent the mean of three biological replicates ± standard deviation. **d**–**f**, Growth curves for *E. coli* varying the gene under the pJEX promoter (**c**): T4 Acb1 (**d**), T4 Acb2 (**e**) and N4 Acb4 (**f**). OD600, optical denisty at 600 nm. **g**, Quantification of DefenseFinder-identified defence systems in the NCBI Reference Sequence Database (RefSeq, 2024; left) versus those within the 10-kb neighbourhood of the Panoptes loci (containing mCpol and 2TMβ). **h**, From the 110 Panoptes loci queried (blue square), the dark green square represents the fraction of CBASS-containing loci nearby Panoptes systems (dark green; containing mCpol and 2TMβ) and unique CD-NTase clusters (light green). **i**, Schematic of the proposed sensing of CBASS interference by the Panoptes_TA defence system. Phage infection with (bottom) and without (top) signalling interference.

proteins (Acb4$_{N4}$ and Acb2$_{T4}$) restored infectivity to levels similar to those observed in the absence of Panoptes (Fig. 4b). Knockdown of Acb1$_{T4}$ partially restored infectivity, consistent with its degradation of the 2'3'-c-di-AMP signalling molecule in vitro (Extended Data Fig. 6d,e). However, knockdown of Acb1$_{T4}$ provided a smaller effect than Acb2$_{T4}$ in the context of T4 infection, potentially reflecting its later gene expression[40]. These results suggest that, in the context of phage infection, Acb expression is necessary for activating Panoptes defence.

To test whether Acb proteins are sufficient to activate Panoptes in the absence of phage infection, we co-expressed Panoptes and candidate activator proteins Acb1$_{T4}$, Acb2$_{T4}$ and Acb4$_{N4}$ (refs. 18–20) (Fig. 4c). We found that low levels of Acb1$_{T4}$ induction[41] were sufficient to activate Panoptes-induced cell death (Fig. 4d), whereas an enzymatically inactivated Acb1$_{T4}$ was insufficient (Extended Data Fig. 6g,h). Similarly, low levels of Acb2$_{T4}$ expression were sufficient to activate Panoptes-induced cell death (Fig. 4e), whereas a known Acb2$_{T4}$-binding site mutant[18] was not (Extended Data Fig. 6i,j). The cyclic nucleotide sponge protein Acb4$_{N4}$ (ref. 20) was also sufficient to activate Panoptes-induced cell death (Fig. 4f and Extended Data Fig. 6k). Collectively, these results demonstrate that Acb signalling interference proteins are necessary and sufficient to activate an immune response from Panoptes.

Antiphage systems tend to colocalize alongside other defence-associated operons in genetic islands[42,43] and can exhibit synergistic activity through either complementary phage resistance patterns or mechanistic traps[44,45]. Given that diverse Acb proteins were both necessary and sufficient to activate Panoptes-induced cell death, we suspected that CBASS systems would be enriched near mCpol-containing genomic loci. To explore this possibility, we analysed all publicly available mCpol-encoding genes and their genomic neighbourhoods, identifying 295 unique mCpol-encoding loci. Phylogenetic analysis of *mCpol* genes highlighted a clade of 109 unique mCpol-encoding loci representing Panoptes (Extended Data Fig. 9, blue highlight). We used Defense-Finder[46], a bioinformatic tool to annotate known prokaryotic antiviral systems, to analyse 109 Panoptes gene neighbourhoods. We identified 91 unique genes encoding CBASS nucleotidyltransferases across 89 of these loci, reflecting a strong enrichment of CBASS relative to the general distribution of defence systems in bacteria[46] (Fig. 4g,h). These CBASS nucleotidyltransferase genes reflected at least 37 unique clusters (80% coverage and 40% identity), suggesting many distinct events of co-association between Panoptes and CBASS (Fig. 4i; examples shown in Extended Data Fig. 9b). We propose that the strong co-occurrence between CBASS and Panoptes reflects a unique detection method for anti-CBASS phage proteins (Fig. 4i). Through inhibitory immune signalling alongside CBASS, Panoptes limits the spread of phages that interfere with CBASS immune signalling.

## Discussion

Our results reveal that the Panoptes locus encodes a defence system composed of a Cas10-like oligonucleotide cyclase (mCpol) and a CBASS-like effector protein (2TMβ) that counters phage-encoded anti-defence mechanisms. mCpol is a constitutively active enzyme that produces c-di-AMP to homeostatically inhibit oligomerization of its 2TMβ effector. The loss of c-di-AMP triggers 2TMβ oligomerization and cell death due to inner membrane collapse, a process activated by phage anti-defence proteins, such as anti-CBASS, which deplete cyclic oligonucleotide messengers[18–20,22,47–49]. When triggered early in the phage infection process, infection aborts due to cell death that protects other cells in a microbial population. These results extend discoveries of virus-encoded anti-immunity proteins[18–20,22,47–49] to show how host systems can counter virus-mediated immune suppression.

We propose that Pantoptes serves as a molecular guard[50] for cyclic nucleotide signalling integrity. By synthesizing a cyclic nucleotide signalling molecule that inhibits cell death by its 2TMβ effector, Panoptes preempts phage-encoded inhibition of CBASS defence systems[3,7]

and/or physiological cyclic di-GMP signalling pathways that impact N4 phage infection[51]. Yet, many genomes containing CBASS systems lack Panoptes, suggesting that its benefits are balanced by potential costs, such as toxicity. Other physiological activities are 'guarded' by bacterial immune systems as well. For example, different retrons appear to guard RecBCD activity or DNA methylation[52,53], whereas the phage anti-restriction-induced system and PrrC-induced abortive infection both sense suppression of restriction-modification defence systems[54–56].

Panoptes represents an inversion of signalling logic compared with currently known CBASS systems, as the 2TMβ effector proteins in each system respond oppositely to small-molecule stimulus. The ability for cyclic nucleotides to either activate or inhibit a cognate receptor mirrors the mechanistic flexibility seen in cyclic di-GMP signalling pathways[57]. Panoptes is, to our knowledge, the first known immune signalling system to exhibit a toxin–antitoxin mechanism of regulation in which the signalling molecule is responsible for effector suppression. The recently characterized Hailong defence system also synthesizes a molecular tripwire to inhibit its cognate effector[58,59]. Instead of synthesizing a cyclic nucleotide, Hailong produces single-stranded DNA oligomers to suppress its effector complex. Phage-encoded exonuclease activity de-suppresses this complex, leading to activation of the effector complex[59]. There are currently eight described RNA or protein antitoxin types, where classification is based on the antitoxin mechanism[31]. Given that mCpol-produced 2'3'-c-di-AMP serves as the antitoxin to the cognate 2TMβ toxin, we propose that Panoptes represents a novel class of TA system (type IX) in which the antitoxin is a small molecule produced by a dedicated antitoxin gene.

Panoptes counters phage anti-defence proteins that promiscuously disrupt cyclic oligonucleotides, acting as a safeguard of immune signalling pathways in antiviral defence. Informatic analysis of closely related Panoptes loci suggests a specific, mechanistic synergy with diverse CBASS systems[44]. This proposed synergy between Panoptes and CBASS immune signalling pathways implies a selective pressure for anti-immune proteins to develop more precise recognition of immune signalling molecules, which probably comes at the cost of broad-spectrum immune evasion. Cross-kingdom mechanisms of immune evasion exert shared evolutionary pressures that suggest similar signalling architectures in eukaryotic immunity are likely to be uncovered[60,61]. Collectively, our results reveal a major constraint for the evolution of viral proteins that inhibit signalling immune systems and suggest the possibility of similar protective roles for cyclic oligonucleotides across the tree of life.

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

# Methods

## Bacterial strains and growth conditions

Standard bacterial cultures were grown in lysogeny broth (LB Lennox) at 37 °C and 250 rpm except where stated otherwise. When necessary, 34 µg ml$^{-1}$ chloramphenicol (+Ch), +50 µg ml$^{-1}$ kanamycin sulfate (+K) or +50 µg ml$^{-1}$ carbenicillin (+C) was supplemented to media and LB agar plates. For cellular assays, all bacterial strains were stored at −80 °C in 25% glycerol (Sigma) when not in use. Cloning and cellular assays were generally performed in *E. coli* DH10b genotype cells (*F- mcrA Δ(mrr-hsdRMS-mcrBC) endA1 recA1 φ80dlacZΔM15 ΔlacX74 araD139 Δ(ara, leu)7697 galU galK rpsL (StrR) nupG λ-*) (Intact Genomics, NEB). For phage experiments involving G17 or Goslar, *E. coli* strain ECOR47 was used[62].

## Phage propagation and scaling

Phages were propagated using standard protocols. Typically, phage production was performed at 37 °C in LB Lennox media infecting *E. coli* BW25113 (*F- DE(araD-araB)567 lacZ4787(del)::rrnB-3 LAM- rph-1 DE(rhaD-rhaB)568 hsdR514*)[63] at an initial MOI of 0.1. Phages G17 and Goslar were propagated on *E. coli* ECOR47. Phage titres were determined on their propagation hosts. Bacteriophages used in this study and their sources[35,36,51] have been described at length in Supplementary File 2.

## Plasmid construction

Plasmids were assembled using PCR, gel extraction (Zymo) and Golden Gate assembly[64] or Gibson Assembly using 25–30 bp of overlapping homology[65]. For some assemblies, Golden Gate ready DNA was ordered from TWIST BioSciences in lieu of generation by PCR. Native promoter Panoptes includes 300 bp upstream of the gene encoding mCpol and 150 bp downstream of the gene encoding 2TMβ. In general, plasmids were propagated in DH10b genotype *E. coli* (Intact Genomics). For G17 and Goslar assays, select plasmids were transferred to *E. coli* ECOR47 through electroporation (Intact Genomics) using standard electroporation parameters (2,000 V and 200 Ω). For protein expression and purification, plasmids were transferred into BL21 AI genotype *E. coli* (*F- ompT hsdSB (rB-mB-) gal dcm araB::T7RNAPtetA*) (Thermo Fisher). For assays involving two plasmids, plasmids were first cloned individually and co-transformed into DH10b genotype *E. coli* (Intact Genomics) through electroporation. All plasmids and co-transformations used in this study were sequenced confirmed by full-plasmid sequencing (Plasmidsaurus). To verify co-transformed plasmids, raw reads were mapped against reference plasmid sequences using Geneious. Plasmids used in this study are listed in Supplementary File 1.

## Phage infection assays

Phage infection plaque assays were performed via double agar overlay. Overlays were formed by mixing 100 µl of saturated overnight cultures grown at 37 °C at 250 rpm with 5 ml molten LB Lennox agar (0.7% w/v, 60 °C). For G17 and Goslar assays, a lower percentage top agar concentration was used (0.35% w/v). The agar–bacterial mixture was supplemented with kanamycin to a final overlay concentration of 50 µg ml$^{-1}$. For plaque assays involving crystal violet (Sigma), crystal violet was supplemented to a final overlay concentration as specified. The top agar and bacterial mixture was poured onto a 5 ml LB agar and kanamycin plate and left to dry under microbiological flame for 15 min. Phages were diluted tenfold in SM buffer (Teknova) and 2 µl of each dilution were spotted onto the top agar overlay and dried under microbiological flame. Once dry, plates were incubated at 30 °C for 12–16 h. Plates were scanned in a standard photo scanner (Epson) and PFU were enumerated, keeping note of changes in plaque size relative to a negative control. 'Lysis from without'[66] phenotypes were interpreted as a lack of productive phage infection and approximated as 1 PFU. Efficiency of plaquing calculations were calculated as mean(PFU in condition)/ mean(PFU negative control). The negative control is defined as a strain harbouring a plasmid encoding red fluorescent protein (RFP) instead of Panoptes (pBA1326). All plaque assays were performed in biological triplicate from independent overnight cultures. Visualizations were performed using Seaborn (v0.13.2) in Python.

For plaque assays involving CRISPRi-ART[39], experiments were performed with the following modifications. Kanamycin, chloramphenicol and aTc (Sigma) were additionally added to final overlay concentrations of 50 µg ml$^{-1}$, 34 µg ml$^{-1}$ and 20 nM, respectively, and plated onto a 5-ml LB agar, kanamycin and chloramphenicol plate. The negative control is defined as Panoptes (pBA1751) alongside a CRISPRi-ART vector with an RFP-targeting guide RNA (pBA635)[39]. Guide sequences are listed in Supplementary File 1. The positive control is defined as a pJEx vector without Panoptes (pBA1326) alongside a CRISPRi-ART guide RNA targeting a phage not present in the experiment. Titres were calculated as mean (PFU condition). All plaque assays were performed in biological triplicate from independent overnight cultures. Visualizations were performed using matplotlib (v3.7.2) and Seaborn (v0.13.2) in Python. CRISPRi-ART guide RNAs were chosen using gRNA1 and gRNA4 designs as previously described[39].

## Bacteriophage liquid growth and phage production

Liquid phage experiments were performed in a Biotek plate reader using LB Lennox + kanamycin media. Strains containing native context Panoptes (pBA1751) or a negative control (pBA1326) plasmid were grown overnight at 37 °C and 250 rpm. Of overnight culture, $8 \times 10^6$ CFU were seeded into each well of a 96-well plate (3903, Corning) in 200 µl media. For phage experiments, T4 was diluted in LB + kanamycin media to achieve defined MOIs during infection except for MOI = 0 in which no phage was added. Growth was monitored in a Biotek Cytation 5 plate reader for 12 h at 800 rpm shaking at 30 °C with OD$_{600}$ readings every 5 min. At the end of the experiment, cultures were pelleted and the supernatant from investigated wells was collected. Phage production was estimated via plaque assay (above) on *E. coli* harbouring a negative control (pBA1326) plasmid and dividing by the effective titre at time 0. All liquid phage experiments were performed in biological triplicate and replicate conditions from independent overnights. Visualizations were performed using matplotlib (v3.7.2) and Seaborn (v0.13.2) in Python.

Estimation of free phage particle production from a single round of infection was performed in 5 ml cultures. Cultures were inoculated with $2 \times 10^8$ CFU *E. coli* harbouring native context Panoptes (pBA1751) or a negative control (pBA1326) plasmid in LB + kanamycin media and incubated at 30 °C at 250 rpm for 15 min. Following incubation, approximately $2 \times 10^6$ CFU of phage T4 was added for a low MOI infection of approximately 0.01. Infections proceeded at 30 °C at 250 rpm and 200 µl sampled every 30 min for 6.5 h. For each 200 µl sample, remaining cells were pelleted, 100 µl supernatant was extracted and stored on ice until every sample was collected. Phage titres were enumerated via plaque assay on *E. coli* harbouring a negative control plasmid (pBA1326). Free phage production was calculated by dividing the sample titre by the number of added phages. Replicates were performed in biological triplicate, sourcing samples from parallel cultures seeded from independent overnights. Visualizations were performed using matplotlib (v3.7.2) and Seaborn (v0.13.2) in Python.

## Panoptes toxin–antitoxin assays

To test for a potential toxin–antitoxin relationship in Panoptes, we cloned candidate antitoxin mCpol or mCpol D57A under control of the pTet promoter in a p15a-CmR plasmid and candidate toxin 2TMβ under control of the pJex promoter[41] in a low-copy SC101-KanR plasmid. dCas13d under pTet control on a p15a-CmR plasmid (pBA635)[39] was used as an antitoxin negative control. RFP expressed under pJex control on a SC101-KanR plasmid was used as a toxin negative control. Both sets of plasmids were co-transformed into DH10b *E. coli* (Intact Genomics) and sequence verified using whole-plasmid sequencing

(Plasmidsaurus), followed by read alignment (Geneious). Sequence-confirmed co-transformants were stored at −80 °C in 20% glycerol (v/v) until further use.

To perform solid agar toxin–antitoxin assays, LB agar plates were freshly poured and dried under flame with the following supplements: 35 µg ml⁻¹ chloramphenicol, 50 µg ml⁻¹ kanamycin, variable amounts of crystal violet to induce toxin expression and variable amounts of aTc to induce antitoxin expression. For toxin expression conditions, +50 nM crystal violet was added. For antitoxin expression conditions, +2 nM aTc was added. Once the agar was dried, three independent overnight cultures containing candidate toxin and antitoxin plasmids were plated in 10× serial dilutions with 5-µl spots and let dry under flame. Once dried, plates were incubated at 30 °C overnight. To let colonies mature for imaging, plates were transferred to a 37 °C incubator for an additional 24 h. Colonies were imaged and counted.

To perform liquid culture toxin–antitoxin assays, three independent overnight cultures containing candidate toxin and antitoxin plasmids were inoculated in LB media at $8 \times 10^6$ CFU in a Corning 3903 plate with the following supplements: 35 µg ml⁻¹ chloramphenicol, 50 µg ml⁻¹ kanamycin, variable amounts of crystal violet to induce toxin expression and variable amounts of aTc to induce antitoxin expression. For toxin expression conditions, +125 nM crystal violet was used. For antitoxin expression conditions, +20 nM aTc was used. The plate was monitored in a Cytation5 plate reader (Biotek) at 30 °C at 807 rpm and $OD_{600}$ measured every 5 min for 12 h. Data were plotted using the Seaborn (v0.13.2) package in Python.

## Panoptes activator assays

To test for an activator relationship with Panoptes, we cloned Panoptes in its native context into a p15a-CmR plasmid and candidate activators T4Acb1, T4Acb1(H44A,H113A), T4Acb2, T4Acb2(Y8A) and N4Acb4 under control of the pJex promoter[41] in a low-copy SC101-KanR plasmid. dCas13d under pTet on a p15a-CmR plasmid (pBA635)[39] was used as the Panoptes negative control. RFP under pJex on a SC101-KanR plasmid was used as an activator negative control.

To perform liquid culture activator assays, three independent overnight cultures containing candidate Panoptes and candidate activator plasmids were inoculated in LB media at $8 \times 10^6$ CFU in a Corning 3903 plate with the following supplements: 35 µg ml⁻¹ chloramphenicol, 50 µg ml⁻¹ kanamycin, variable amounts of crystal violet to induce candidate activator expression and no aTc for Panoptes expression. For candidate activator expression conditions, +125 nM crystal violet was used. The plate was monitored in a Cytation5 plate reader (Biotek) at 30 °C, 807 rpm and $OD_{600}$ measured every 5 min for 12 h. Data were plotted using the Seaborn (v0.13.2) package in Python.

To measure viability from liquid culture activator assays, we used a modified protocol. Three independent overnight cultures containing candidate pTet-native promoter Panoptes or pTet-dCas13d (negative control) and pJEX-T4Acb2 or pJEX-RFP (negative control) were inoculated in LB media at $8 \times 10^6$ CFU in a Corning 3903 plate with the following supplements: 35 µg ml⁻¹ chloramphenicol, 50 µg ml⁻¹ kanamycin and no inducers. Cells were grown for 3 h in a Cytation5 plate reader (Biotek) at 30 °C and 807 rpm ($OD_{600}$ ~ 0.2). At this point ($t = 0$), either +125 nM crystal violet or +0 nM was supplemented to the cultures to activate or not activate Panoptes and grown for an additional 2 h ($t = 120$). At both $t = 0$ and $t = 120$, 5 µl of cells were sampled, tenfold serially diluted in inducer-free media and spotted on inducer-free LB agar with antibiotics. Plates were incubated overnight at 37 °C and colonies counted to estimate viability from $t = 0$ and $t = 120$. Data were plotted using the Seaborn (v0.13.2) package in Python.

## Protein expression and purification

Expression sequences for ECOR31 mCpol were cloned into a custom pET-based vector by Gibson assembly to yield an N-terminal His₁₀-MBP-TEV or C-terminal TEV-MBP-His₁₀ construct. Expression plasmids for ECOR31 TM2β residues 72–200 (2TMβ(Δ2TM)) were cloned into a pET Duet-1 vector for co-expression of His₆-SUMO2-TM2β and MBP-TEV-mCpol constructs.

Proteins were expressed in *E. coli* Rosetta 2 (DE3) pLysS by growing cells to an $OD_{600}$ of 0.4–0.6 in 2× yeast extract tryptone (2×YT) medium at 37 °C and induced with 0.5 mM isopropyl β-ᴅ-1-thiogalactopyranoside (IPTG) following a cold shock at 4 °C. After induction, cells expressing each protein were grown overnight at 16 °C. Cells were collected by centrifugation for 20 min at 12,300*g* and 4 °C and resuspended in 20 mM Tris-HCl, pH 8.0, 10 mM imidazole, 2 mM MgCl₂, 500 mM KCl, 10% (v/v) glycerol, 0.5 mM Tris (2-carboxyethyl) phosphine and Roche protease inhibitor.

Cells were lysed by sonication, and cell lysate was clarified by centrifugation at 17,000*g* and 4 °C for 0.5 h. The supernatant was bound to Nickel-NTA affinity resin pre-equilibrated with wash buffer (20 mM Tris-HCl, pH 8.0, 500 mM KCl, 30 mM imidazole, 10% (v/v) glycerol and 0.5 mM Tris(2-carboxyethyl) phosphine (TCEP)) for 1 h at 4 °C. Supernatant was discarded and resin was washed 5 × 30 ml wash buffer (20 mM Tris-HCl, pH 8.0, 500 mM KCl, 30 mM imidazole, 10% (v/v) glycerol and 0.5 mM TCEP). All buffers for the purification of ECOR31 TM2β were additionally supplemented with 1 µM 2′3′-c-di-AMP (BioLog Life Science Institute) in the washing and subsequent steps. Protein was eluted in 5 ml elution buffer (20 mM Tris-HCl, pH 8.0, 500 mM KCl, 300 mM imidazole, 10% (v/v) glycerol and 0.5 mM TCEP). Recombinant TEV protease or SUMO protease 2 (SENP2) with an N-terminal His-tag (BPS Bioscience) was added to the elution for cleavage and dialysed overnight at 4 °C in a 3.5 kDa MWCO dialysis cassette (Thermo Fisher Scientific) with dialysis buffer (20 mM Tris-HCl, pH 8.0, 500 mM KCl, 30 mM imidazole, 10% (v/v) glycerol and 0.5 mM TCEP). The resultant solution was passed over a 5-ml Ni-NTA Superflow cartridge (Cytiva). Flow-through was collected, concentrated to less than 2 ml using a 3 kDa MWCO centrifugal filter (Amicon), and loaded onto a HiLoad 16/600 Superdex 200-pg column (Cytiva). Elution was isocratic (20 mM Tris-HCl, pH 8.0, 500 mM KCl, 10% glycerol and 1 mM TCEP) and monitored by absorbance at 280 nm. Peaks were pooled (approximately 86 ml for ECOR31 TM2β; approximately 83 ml for ECOR31 mCpol), concentrated to 5 mg ml⁻¹ (ECOR31 TM2β) or 3 mg ml⁻¹ (ECOR31 mCpol) as determined by absorbance at 280 nm, snap frozen and stored at −80 °C.

## Analysis of recombinant protein

Purified protein was analysed by SDS–PAGE. Samples were prepared in 1X protein loading dye (50 mM Tris-HCl, pH 6.8, 15 mM EDTA, 6% (v/v) glycerol, 10% SDS and bromophenol blue), heated at 95 °C for 3 min, loaded onto a 12% Mini-Protean TGX precast protein gel (Bio-Rad) and run at 125 V until the dye front reached the bottom of the gel. Gels were stained in 30% ethanol, 10% glacial acetic acid in water and 0.1% (w/v) R-250 Coomassie, and destained in 40% ethanol and 10% glacial acetic acid in water.

## Crystallization and structure determination

Crystals of mCpol bound to a hydrolysis-resistant amine-modified analog of ATP (ApNHpp; NU-449-1, Sapphire North America) were grown at 20 °C using the hanging drop vapour diffusion method. A 1-µl solution of 3 mg ml⁻¹ mCpol in 20 mM Tris-HCl, pH 8.0, 500 mM NaCl and 5% glycerol was mixed with 1 µl 0.1 M bicine, pH 8.0, and 15% PEG 1500 and supplemented with 0.2 µl 10 mM MgCl₂ with 10 mM of ApNHpp. Single crystals appeared within 3 days and were cryoprotected in a solution of mother liquor with 30% glycerol before being flash cooled in liquid nitrogen.

Data for mCpol bound to ApNHpp were collected via fine-phi slicing using 0.2° oscillations at beamline 12-2 at Stanford Synchrotron Radiation Lightsource at SLAC National Accelerator Laboratory. X-ray diffraction data were measured to 2.28 Å resolution.

## Processing and refinement of crystallographic data

Crystallographic data were processed with the SSRL autoxds script with an I/sigI cut-off ≥ 1.50. Crystals displayed moderate anisotropic X-ray diffraction with some diffraction extending beyond 2.0 Å resolution. However, the resolution was isotropically truncated to 2.28 Å resolution to generate a robust complete dataset. The structure was solved by molecular replacement[67] using a ColabFold-generated[68] model (pLDDT = 95.12) of mCpol with residual residues from the C-terminal cleavage site (mCpol-ENLYFQ) in PHENIX[69]. Molecular replacement successfully identified the placement of two mCpol monomers as indicated by the log-likelihood gain of 556.03 and the translation-function $Z$ score of 23.6. The structure was refined using PHENIX[70] including simulated annealing, non-crystallographic symmetry and TLS (translation, libration, screw-rotation) parameters. In each protein monomer (chains A and B) residues 121–125 were disordered and thus not included in the model. The model was built and adjusted using COOT[71]. The structure was refined to a final $R_{free}$ and $R_{work}$ of 24.88% and 22.43%, respectively (Extended Data Table 1). Extended Data Table 1 shows data processing and model refinement statistics. Atomic coordinates and structure factors have been deposited to the Protein Data Bank.

## Sequence alignment of mCpol domains

Proteins in the Pfam entry PF18182 'minimal CRISPR polymerase domain' were downloaded and used to create an alignment of 163 mCpol sequences. Sequences were trimmed and aligned using MAFFT alignment with default parameters in Geneious Prime (v2023.2.1).

## TLC of cyclase products

Recombinant enzymes were assessed for cyclase activity by in vitro reactions with nucleoside triphosphates and analysis by TLC. Cyclase activity assays were initiated by the addition of recombinant enzyme (40 μM final) in reaction buffer (50 mM Tris, pH 8.0, 10 mM $MgCl_2$ and 100 mM NaCl) to 5 mM NTP (Thermo Scientific). The reaction mixture was incubated at 37 °C for 18 h and stopped by vortexing for 20 s.

Recombinant enzymes were assessed for cyclic dinucleotide (c-di-AMP) degradation in biochemical reactions containing buffered c-di-AMP and products were analysed by TLC. Reactions were initiated by the addition of recombinant enzyme (40 μM) in reaction buffer (50 mM Tris, pH 8.0, 10 mM $MgCl_2$ and 100 mM NaCl) to 1.25 mM c-di-AMP (Biolog or MedChemExpress). The reaction mixture was incubated at 37 °C from 10 min to 18 h and stopped by vortexing for 20 s.

To silica gel 60 matrix TLC plates (5 cm × 10 cm, glass support with fluorescent indicator 254 (Supelco)), was added 2 μl in vitro enzymatic reaction mix. Separation was performed in an eluent of n-propanol:ammonium hydroxide:water (11:7:2 v/v/v) for 45 min. The plate was allowed to dry fully and visualized with a short-wave ultraviolet light source at 254 nm. Uncropped TLC images are available in Supplementary Fig. 1.

## S1 nuclease digestion of cyclase products

To determine the linkage identity of products formed in vitro, cyclase reactions described previously were subjected to digestion by S1 nuclease. To a reaction containing 1X S1 nuclease buffer (40 mM sodium acetate (pH 4.5 at 25 °C) and 300 mM NaCl₂), 20 μl of the vortex-inactivated cyclase reaction and 200 U S1 nuclease (Thermo Scientific) were added to a total volume of 40 μl. The reaction was incubated at 37 °C for 4 h and inactivated by the addition of 2 μl of 0.5 M EDTA with heating (70 °C for 10 min).

## Preparation of extracts for mass spectrometry

mCpol–ATP reactions were diluted 1:2 in deionized H₂O, centrifuged at 13,000g for 15 min and used directly in subsequent analysis. Species identified by TLC were directly analysed by TLC–MS. Silica containing the product was scraped away from the TLC plate and added to 40 μl water. The resultant slurry was vortexed and heated at 30 °C for 10 min and then centrifuged at 13,000g for 15 min. The supernatant was removed for subsequent analysis.

## Cyclase product analysis by HPLC–MS

Cyclase product extracts were analysed using a liquid chromatography system (1200 series, Agilent Technologies) that was connected in line with an LTQ-Orbitrap-XL mass spectrometer equipped with an electrospray ionization source (Thermo Fisher Scientific). The liquid chromatography system was equipped with a G1322A solvent degasser, G1311A quaternary pump, G1316A thermostatted column compartment and G1329A autosampler unit (Agilent). The column compartment was equipped with an Ultra C18 column (length of 150 mm, inner diameter of 2.1 mm and particle size of 3 μm; 9174362, Restek). Ammonium acetate (98% or more, Sigma-Aldrich), methanol (Optima LC–MS grade, 99.9% minimum; Fisher) and water purified to a resistivity of 18.2 MΩ cm⁻¹ (at 25 °C) using a Milli-Q Gradient ultrapure water purification system (Millipore) were used to prepare mobile-phase solvents. Mobile-phase solvent A was water and mobile-phase solvent B was methanol, both of which contained 10 mM ammonium acetate. The elution program consisted of isocratic flow at 0.5% (v/v) B for 2 min, a linear gradient to 99.5% B over 2 min, isocratic flow at 99.5% B for 4 min, a linear gradient to 0.5% B over 1 min, and isocratic flow at 0.5% B for 21 min, at a flow rate of 100 μl min⁻¹. The column compartment was maintained at 30 °C and the sample injection volume was 1 μl. External mass calibration was performed in the positive ion mode using the Pierce LTQ electrospray ionization positive ion calibration solution (88322, Thermo Fisher Scientific). Full-scan, high-resolution mass spectra were acquired in the positive ion mode over the range of $m/z$ = 300–2,000, using the Orbitrap mass analyzer, in profile format, with a mass resolution setting of 60,000 (at $m/z$ = 400, measured at full-width at half-maximum peak height). Tandem mass (MS/MS or MS²) spectra were acquired using collision-induced dissociation in the linear ion trap, in centroid format, with the following parameters: isolation width = 3 $m/z$ units, normalized collision energy = 28%, activation $Q$ = 0.25 and activation time = 30 ms. Data acquisition was controlled using Xcalibur software (v2.0.7, Thermo Fisher Scientific).

## Mass spectrometry data processing

Raw data were converted to mzXML format using msconvert (v3.0.19052.1) from the Galaxy platform[72]. Data were then processed using the open source software MZmine (v3.9.0). Compound identification was performed by differential mass spectrometry of in vitro reactions with active and inactive enzymes, and by comparing retention times and $m/z$ with those of chemical standards.

## 2′3′-c-di-AMP titration

Recombinant 2TMβ(Δ2TM) was dialysed overnight at 4 °C into SEC buffer (20 mM Tris-HCl, pH 8.0, 500 mM KCl, 10% glycerol and 1 mM TCEP). Protein samples were diluted to 4.75 mg ml⁻¹ (323 μM) in SEC buffer with varying concentrations of 2′3′-c-di-AMP (0, 500 μM, 1 mM and 2 mM) and 100 μl was loaded into a Superdex 200 Increase 3.2/300 column (Cytiva). Elution was isocratic in SEC buffer and monitored by A280. Data were visualized in GraphPad Prism (v10.2.2).

## Thermal shift assay of 2TMβ

Spectra were obtained using a Bio-Rad CFX Connect Real-Time PCR Detection System. Solutions contained 1 μM 2TMβ(Δ2TM) in 20 mM Tris-HCl, pH 8.0, 500 mM KCl, 10% glycerol, 1 mM TCEP and 2X SYPRO orange dye with or without 20 μM 2′3′-c-di-AMP. To a 96-well half-skirted clear-bottom PCR plate (Axygen) was added 20 μl of each solution. Wells were sealed with Axyseal sealing film (Axygen). Fluorescence was measured as the solutions were heated from 18 °C to 90 °C at a rate of 2 °C per minute. The fluorescence signal as a function of temperature was plotted, and the background values of buffered solution (with

or without 2′3′-c-di-AMP) without protein was subtracted from each sample. Measurements were performed in triplicate and the graph is the average of three replicates ± standard deviation.

## Native PAGE

Recombinant 2TMβ(Δ2TM) was dialysed overnight at 4 °C into 20 mM Tris-HCl, pH 8.0, 500 mM KCl, 10% glycerol and 1 mM TCEP. To 1X native loading buffer (50 mM Tris-Cl, pH 8.0, 0.1% bromophenol blue, 10% glycerol and 100 mM dithiothreitol) 4 µg 2TMβ(Δ2TM) was added at 4 °C. The sample was analysed by electrophoresis on an 8% native polyacrylamide gel (5% 29:1 acrylamide:bis solution, 750 mM Tris-Cl, pH 8.8) in Tris-glycine running buffer (1.5% w/v Tris-Cl, pH 8.8, and 9.4% (w/v) electrophoresis grade glycine) at 4 °C and 150 V for 105 min. The gel was stained in 30% ethanol, 10% glacial acetic acid in water and 0.1% (w/v) R-250 Coomassie, and destained in 40% ethanol and 10% glacial acetic acid in water. Uncropped gel images are available in Supplementary Fig. 1.

## Fluorescence microscopy

Inducing agents or water were added to 400 µl or 200 µl of log phase cultures at $OD_{600}$ 0.12–0.2 (final concentrations: aTc = 200 nM and crystal violet = 50 nM), which were then incubated in a roller for 50 min at 30 °C. Stains were added to the cultures and incubated, rolling, for an additional 10 min at 30 °C. The final stain concentrations were 4 µg ml$^{-1}$ DAPI and either 2.5 µg ml$^{-1}$ MitoTracker Green FM (Fig. 3i–k) or 1 µg ml$^{-1}$ FM4-64 and 2.5 µg ml$^{-1}$ SYTOX Green (Extended Data Fig. 6). FM4-64 is a membrane stain and was used to verify the presence of membrane lesions in the SYTOX Green staining assay, but is not shown. Of culture, 20 µl was spotted on an agarose imaging pad (25% LB and 1% agarose) in single-well concavity slides for imaging. Cells were imaged at room temperature with eight 0.2-µm slices in the z axis. Images were collected on a DeltaVision Elite Deconvolution microscope (Applied Precision) using DeltaVision SoftWoRx (v6.5.2) and figure panels were prepared in Adobe Photoshop (v21.2.0) and Adobe Illustrator (v24.2).

## Sequence mining and genomic neighbourhood analysis

A collection of mCpol encoding genes was curated using the following strategy. First, a DALI search was performed against representatives of the clustered AlphaFold Database (AFDB) using an AlphaFold2 model of the mCpol (A0A426EXS8). After filtering for a Z score of 12, a structurally informed multiple sequence alignment (MSA) was generated and used for phylogenetic analysis as previously described[73]. The clade containing mCpol representatives (n = 5) was extracted from the resulting tree, and the selection was then expanded to include all cluster members (n = 29). A structurally informed MSA was generated from this set and used as input for an HMM search against the NCBI_NR_Aug_2018 database on the MPI Bioinformatics Toolkit website (MSA enrichment iterations using HHblits: 1, e-value cut-off for reporting: e−10, maximum target hits: 1,000). This set of mCpol candidates (n = 592) was aligned using MAFFT (v7.490) in Geneious Prime (v2023.0.1; algorithm: E-INS-i, scoring matrix: BLOSUM62 and gap open penalty: 1.3). Fragmented mCpol genes were removed from the alignment, and the remaining genes were used for neighbourhood analysis (n = 593).

A custom Python script was used to retrieve Genbank files of mCpol-containing loci. Only contigs longer than 20 kb were kept for further analysis (n = 505). The resultings contigs were converted to FASTA format and deduplicated using mmseqs[74] easy-cluster (-c 0.5 --min-seq-id 0.5 --dbtype 2 --cluster-mode 2). The final set of sequences (n = 252) were then subjected to defence association analysis via Defense-Finder Web Service (v2.0.0)[46,75] (https://defensefinder.mdmlab.fr/. The outputs of DefenseFinder were then further processed using a custom Python script to add defence system hit annotations to the Genbank files. This final output was used to investigate co-occurrence of mCpol with CBASS and other types of defence systems.

## Taxonomic analysis of mCpol-encoding genes

To determine the taxonomic distribution of mCpol homologues, we retrieved the NCBI taxonomic identifier (taxID) associated with each protein sequence. Protein IDs were supplied to a custom script that queries the NCBI Entrez system via E-utilities (esearch, elink, efetch and xtract)[76] to extract both the taxID. For protein IDs that failed automated retrieval, taxonomic information was manually obtained using NCBI's web interface. For each taxID, we used a separate custom Python script that calls the NCBI Datasets[77] command-line tool to obtain taxonomy summaries in JSON format and subsequently parses the taxonomic ranks saved as a tsv. A Jupyter notebook was used to further cross-referenced protein IDs with locus tag information from associated annotation files. Merging and reformatting of the dataset was performed in Python using pandas, yielding a finalized tab-delimited table containing accession IDs, taxonomic lineages and protein identifiers for downstream analysis.

## Reporting summary

Further information on research design is available in the Nature Portfolio Reporting Summary linked to this article.

## Data availability

Plasmids and plasmid sequences are available on Addgene under the article title (Addgene no. 246058-246067). The structure of mCpol–ApNHpp has been deposited under PDB ID 9NWN and is publicly available. Structures accessed are publicly available under PDB ID 6IFK, 6IFL and 3W2W. Mass spectrometry source data (Fig. 2b and Extended Data Fig. 2c–f) have been deposited at Figshare project number 261113 (https://figshare.com/projects/A_miniature_CRISPR-Cas10_enzyme_confers_immunity_by_inhibitory_signaling/261113). Fluorescence microscopy images have been deposited at Figshare project number 260219 (https://figshare.com/projects/A_miniature_CRISPR-Cas10_enzyme_confers_immunity_by_an_inhibitory_signaling_pathway/260219) (Fig. 3i–k and Extended Data Fig. 4a–c). Taxonomic analysis source data (Extended Data Fig. 9a) are available through Zenodo (https://zenodo.org/records/16898616). Source data are provided with this paper.

## Code availability

All scripts used to perform the analyses are publicly available through Zenodo (https://zenodo.org/records/16898616). Scripts for taxonomic analysis are available on GitHub (https://github.com/kenloi/mpcol_taxa).

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

**Acknowledgements** We thank members of the Doudna laboratory and collaborators including D. Colognori, J. Nomburg, O. Tuck, D. Bellieny, L. Chen, A. J. Fisher, P. D. Adams, I. Esain-Garcia and S. Lopez for discussion, encouragement and feedback; P. Skopinstev for advising on protein crystallization and for assisting with crystal mounting; and K. Lucas for her exceptional operational and administrative leadership in support of the lab's research and activities. J.A.D. is an investigator of the Howard Hughes Medical Institute (HHMI). Research in the Doudna laboratory is supported by the HHMI, the US National Institutes of Health (NIH)/National Institute of Allergy and Infectious Diseases (NIAID; U19AI171110, U54AI170792, U19AI135990, UH3AI150552 and U01AI142817), NIH/National Institute of Neurological Disorders and Stroke (U19NS132303), NIH/National Heart, Lung, and Blood Institute (R21HL173710), National Science Foundation (2334028), US Department of Energy (DE-AC02-05CH11231, 2553571 and B656358), Lawrence Livermore National Laboratory, Apple Tree Partners (24180), L. K. Shing, Koret-Berkeley-TAU, Emerson Collective and the Innovative Genomics Institute. The authors acknowledge funding from the HHMI Emerging Pathogens Initiative grant (to J.P.) and NIH grant R01-GM129245 (to J.P.). E.E.D. was supported by the National Institute of General Medical Sciences (NIGMS) of the NIH under award number F32GM153031. B.A.A. was supported by m-CAFEs Microbial Community Analysis and Functional Evaluation in Soils (m-CAFEs@lbl.gov), a science focus Area led by Lawrence Berkeley National Laboratory based on work supported by the US Department of Energy, Office of Science, Office of Biological and Environmental Research under contract number DE-AC02-05CH11231. P.H.Y. was supported by a National Science Foundation Graduate Research Fellowship. E.G.A. was supported by an NIH PiBS training grant (T32 grant GM133351). C.S.B. was supported by a University of California Office of the President funded UC-Historically Black Colleges and Universities Initiative (UC-HBCU) award to the Doudna laboratory. X.E.W. was supported by NIAID of the NIH under award number T32AI145821. Use of the Stanford Synchrotron Radiation Lightsource, SLAC National Accelerator Laboratory, is supported by the US Department of Energy, Office of Science, Office of Basic Energy Sciences under contract no. DE-AC02-76SF00515. The SSRL Structural Molecular Biology Program is supported by the US Department of Energy Office of Biological and Environmental Research, and by the NIGMS of the NIH (P30GM133894). The QB3/Chemistry Mass Spectrometry Facility received NIH support (grant number 1S10OD020062-01). The contents of this publication are solely the responsibility of the authors and do not necessarily represent the official views of the NIGMS or the NIH.

**Author contributions** E.E.D., B.A.A. and J.A.D. conceived the project. E.E.D., B.A.A., K.H., E.G.A., A.L., C.S.B. and A.A. performed the experiments. E.E.D. led biochemical and structural experiments and analyses. B.A.A. led microbiological and genetics experiments and analyses. B.A.A., P.H.Y. and K.L. performed the bioinformatics experiments and analyses. E.G.A. performed the microscopy experiments. E.E.D. and X.E.W. processed and refined the crystallographic data. A.T.I. collected the mass spectrometry data. J.P. supervised the microscopy experiments. E.E.D., B.A.A. and J.A.D. wrote the original draft of the manuscript. All authors edited the manuscript and support its conclusions.

**Competing interests** J.A.D. is a co-founder of Caribou Biosciences, Editas Medicine, Intellia Therapeutics, Scribe Therapeutics, Mammoth Biosciences, Evercrisp and Azalea; a scientific advisory board member of Caribou Biosciences, Scribe Therapeutics, Mammoth and Inari; and a Director at Johnson & Johnson, Tempus and Altos Labs. J.P. has an equity interest in Linnaeus Bioscience Incorporated and receives income. The terms of this arrangement have been reviewed and approved by the University of California, San Diego in accordance with its conflict-of-interest policies. The other authors declare no competing interests.

**Additional information**
**Correspondence and requests for materials** should be addressed to Jennifer A. Doudna.

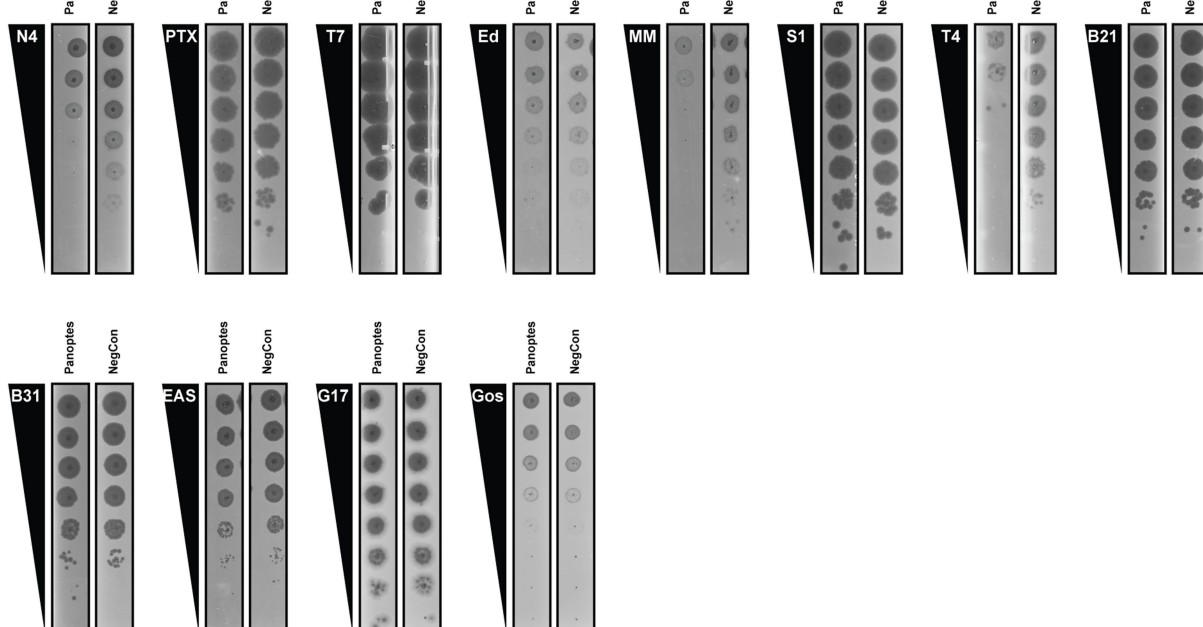

**Extended Data Fig. 1 | Representative plaque assays for phage sensitivity to Panoptes.** Plaque assays of phages shown in the order they appear in Fig. 1 and their sensitivity to ECOR31 Panoptes-mediated antiphage defense. Plaque assays shown are representative images from 3 biological replicates. NegCon is a strain harboring a plasmid encoding red fluorescent protein (RFP) 497 instead of Panoptes (pBA1326, Supplementary File 1).

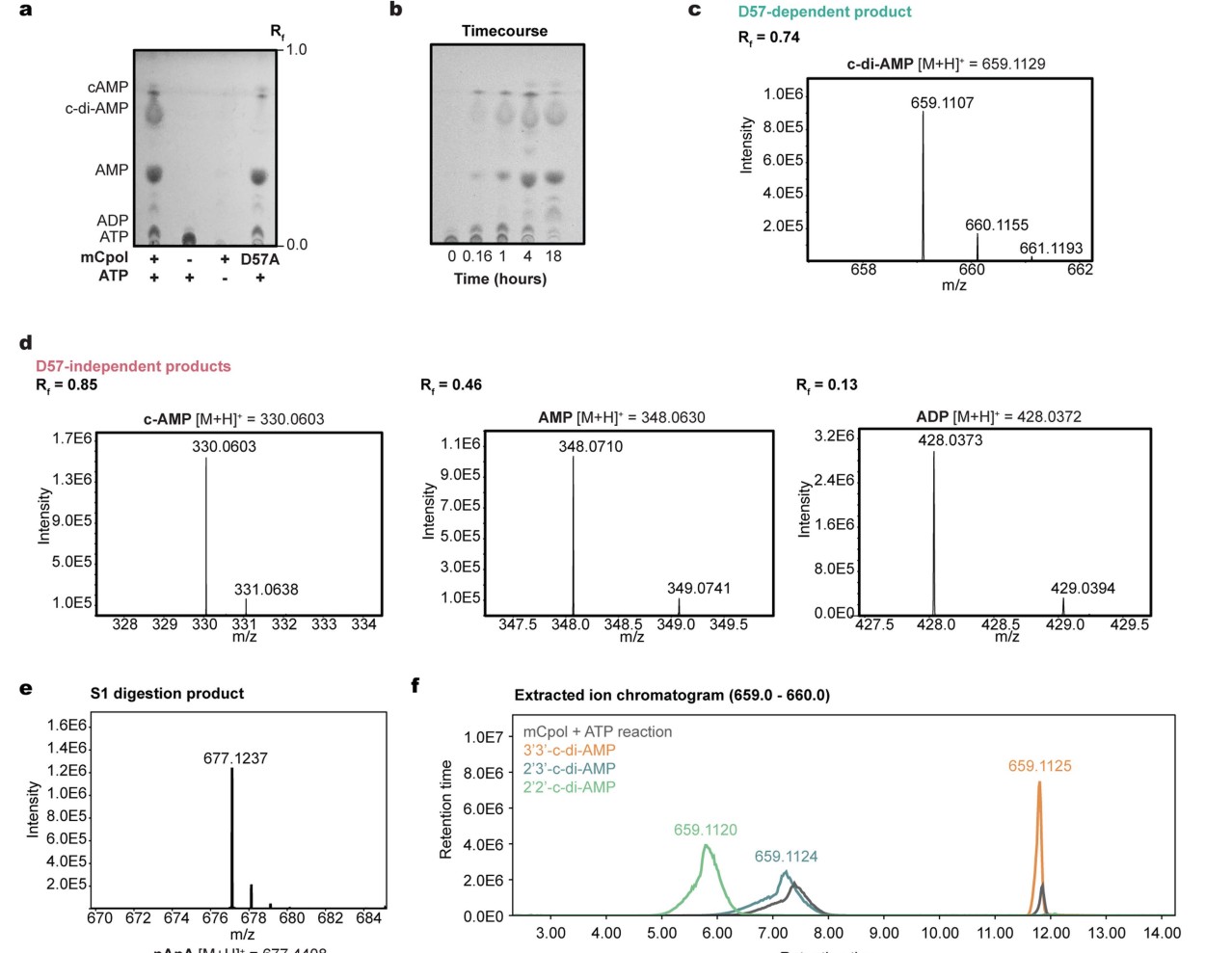

**Extended Data Fig. 2 | Thin Layer Chromatography-Mass Spectrometry (TLC-MS) characterization of products formed when recombinant mCpol is added to ATP.** (a) A representative TLC plate showing reaction between mCpol or mCpol D57A and ATP by $R_f$. (b) Time course of mCpol-ATP reaction from 10 min to 18 h. (c) MS of the mCpol D57-dependent product spot. (d) MS of the 3 major product spots that are not dependent upon the conserved aspartate in mCpol. (e) Mass chromatogram of S1 digestion of the major reaction product, which corresponds to the molecular weight of linear pApA. (f) Extracted mass chromatogram ($m/z$ 659-660) for c-di-AMP standards with linkage identities 2'3', 3'3', and 2'2' versus the mCpol-ATP reaction.

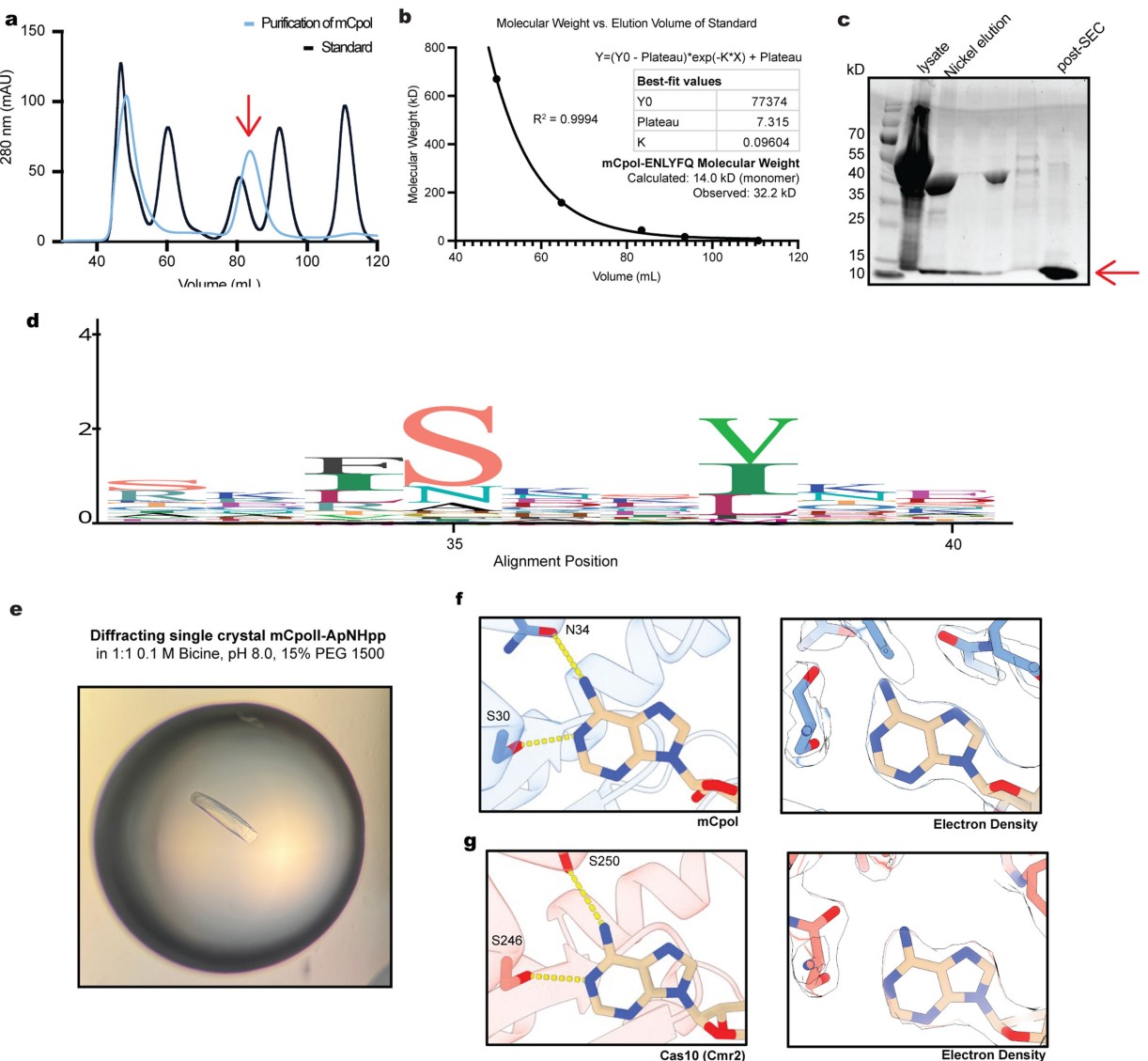

**Extended Data Fig. 3 | Crystallization of ECOR31 mCpol and mechanism of adenine base specificity.** (a) Size exclusion chromatography (SEC) of ECOR31 mCpol with ENLYFQ cleavage site (red arrow) versus a molecular weight standard. MW of standard peaks in kD: 670, 158, 44, 17, 1.35). (b) Determination of mCpol multimerization state by comparison with a molecular weight standard. The peak at ~48 mL represents the void volume of the column and the peak at ~83 mL (red arrow) is a dimer of mCpol. (c) Purification of mCpol for crystallography (trays set up with post-SEC fraction, red arrow). (d) Conservation of residues governing base specificity in an alignment of 163 mCpol domain-containing proteins. Alignment position 35 equates to S30 and position 39 equates to N34 in ECOR31 mCpol. (e) Single crystal resulting from the 0.1 M bicine, pH 8.0, 15% PEG 1500 condition, mixed 1:1 with protein and supplemented with 0.2 equivalents of 10 mM of ApNHp in 10 mM MgCl$_2$. (f) Base specific contacts made between mCpol and the adenine base and corresponding electron density. (g) Base specific contacts made between Cas10 (Cmr2, PDB ID: 3W2W) and the adenine base and corresponding electron density.

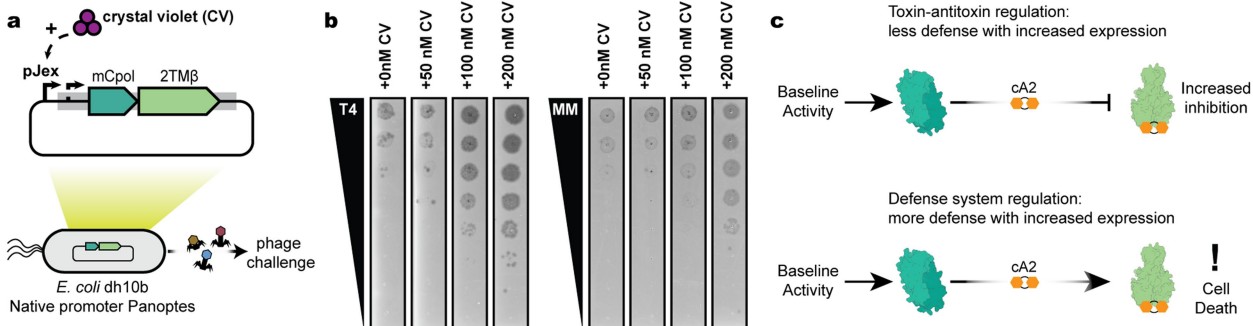

**Extended Data Fig. 4 | Increasing expression of Panoptes yields decreasing levels of phage defense.** (a) Overview of phage infection experiment with additional mCpol induction. (b) Plaque assays of T4 and MM02 phages in the presence of increasing expression levels of Panoptes at specified concentrations of CV. Plaque assays are representative images of 3 biological replicates.

(c) Comparison of toxin-antitoxin (top) and defense system architectures (bottom). Increased expression of a toxin-antitoxin signaling system would also yield increased inhibition of the toxin and less defense. Increased expression of a defense system would yield higher levels of defense and potentially incidental cell death.

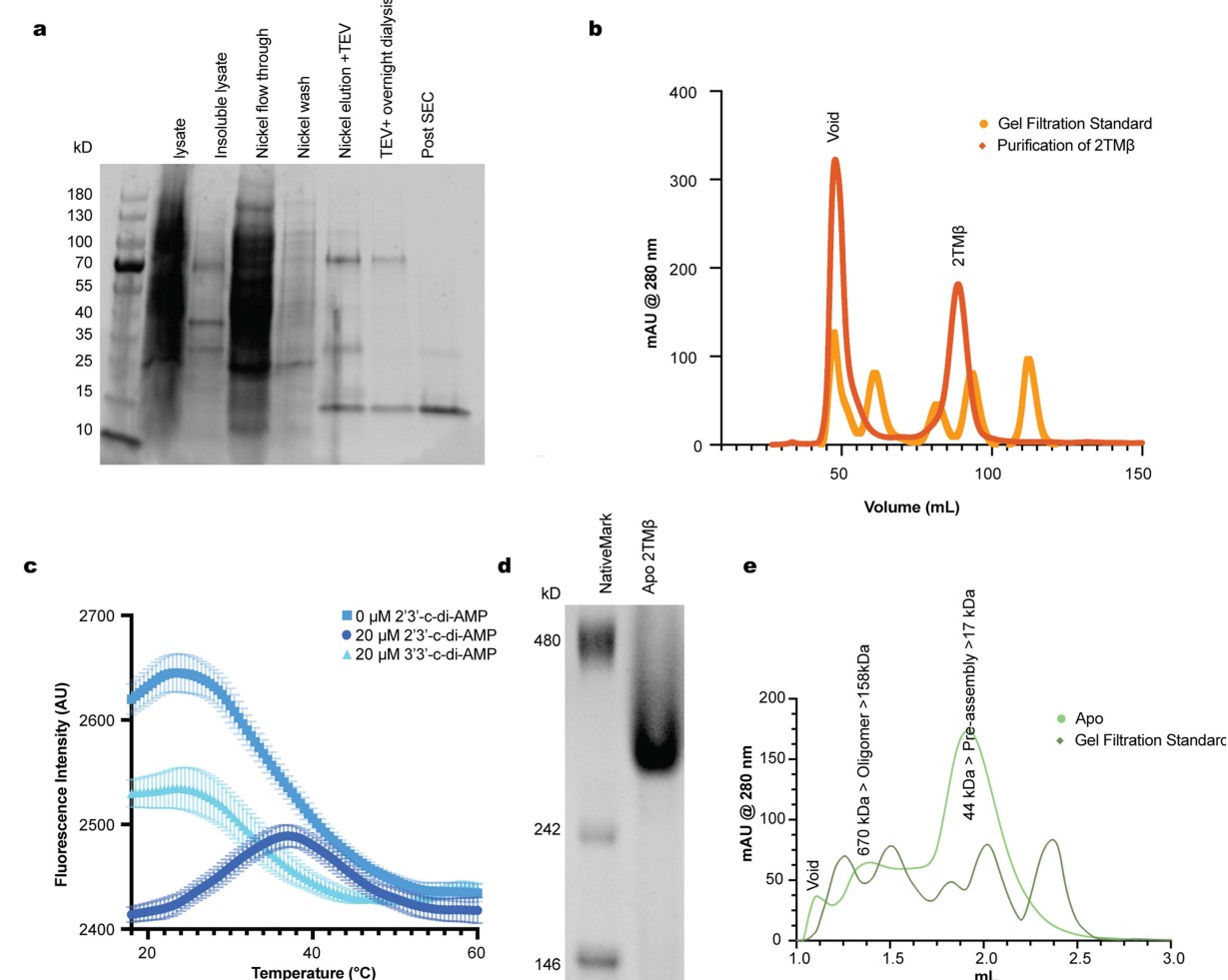

**Extended Data Fig. 5 | Characterization of 2TMβ from ECOR31.**
(a) Denaturing PAGE gel of fractions from purification of 2TMβ Δ2TM versus PageRuler Prestained Protein Ladder 10 to 180 kD. Fractions are labeled above each well. The Post SEC fraction was used for further analysis. (b) mAU (milli Absorbance Units) over time for SEC purification of 2TMβ Δ2TM on a HiLoad 16/600 Superdex 200 pg column as compared to a Gel Filtration Standard (BioRad). MW of standard peaks in kD: 670, 158, 44, 17, 1.35. Void volume and 2TMβ Δ2TM peak collected (corresponding to Post SEC in (a)) are labeled. (c) Thermal melting assay measuring SYPRO orange fluorescence intensity in arbitrary units (AU) as temperature is increased for samples of 1 μM 2TMβ Δ2TM with 0 or 20 μM cyclic dinucleotide. SYPRO orange binds denatured protein or hydrophobic patches and results in an increase in fluorescence upon protein denaturation. The high initial fluorescence of the apo form indicates that the protein is partially unfolded under this condition. Partial stabilization is observed with the addition of 3'3'-c-di-AMP, while the addition of 2'3'-c-di-AMP leads to more complete stabilization by a shift in the apparent melting temperature. (d) Native PAGE gel of the the 44 kDa peak from SEC of 2TMβ Δ2TM (e) compared to the Native Mark Unstained Protein Ladder (Thermo). (e) Analytical SEC of purified 2TMβ Δ2TM versus a Gel Filtration Standard (MW of standard peaks in kD: 670, 158, 44, 17, 1.35) on a Superdex 200 Increase 3.2/300 column. Apo data shown is the same as Fig. 3f for comparison. Pre-assembly state and oligomer 2TMβ Δ2TM peaks are labelled.

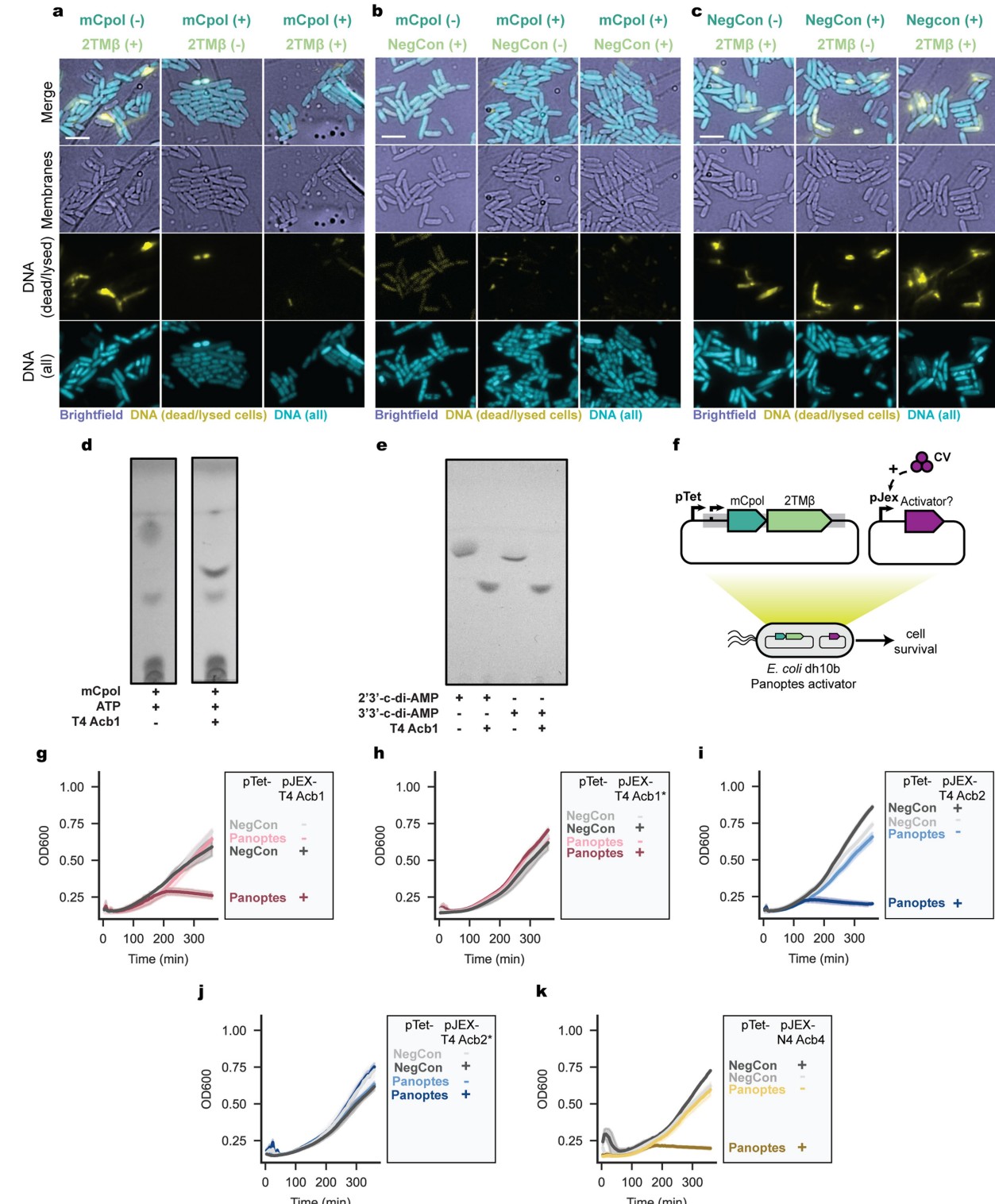

**Extended Data Fig. 6** | See next page for caption.

**Extended Data Fig. 6 | mCpol reaction product suppresses 2TMβ.** (a-c) SYTOX green assay. Induction (+) or lack of induction (−) for mCpol, 2TMβ, and vector controls. (a) Fluorescence microscopy shows cell lysis (via SYTOX Green stain) if 2TMβ is expressed in the absence of mCpol. (b) Fluorescence microscopy shows no or low background cell lysis for a 2TMβ vector control. (c) Fluorescence microscopy cell lysis for leaky (-) and induced (+) expression of 2TMβ in a strain with a vector control lacking mCpol. Scale bars = 5 μm. (d) Treatment of mCpol reaction products with T4 Acb1. mCpol reaction before and after treatment with T4 Acb1 shows disappearance of the c-di-AMP product. (e) Digestion of 2'3'-c-di-AMP and 3'3'-c-di-AMP with T4 Acb1. (f-k) Mutant controls for Panoptes Activator assays. (f) Overview of activator assays in the context of Panoptes under control of its native promoter. For activator assays in panels b-f candidate Panoptes activator proteins are expressed at +0 or +125 nM CV. (g) T4 Acb1 expression is sufficient to activate Panoptes toxicity. Each condition contains two plasmids. The left column indicates the identity of the plasmid under pTet control, which is either a negative control (NegCon) plasmid or an otherwise identical plasmid containing the Panoptes system. The right column indicates whether the second plasmid, containing T4 Acb1 under the control of a pJEx promoter has been induced (+) at 125 nM crystal violet (CV) or not induced (−). (h) Catalytically deactivated mutant of T4 Acb1 (H44A, H113A) expressed is insufficient to activate Panoptes toxicity. (i) T4 Acb2 expression is sufficient to activate Panoptes toxicity. Plasmid identities follow those in 4 d, where T4 Acb2 is now under the control of the pJEx promoter. (j) Binding-deficient mutant of T4 Acb2 (Y8A) expression is insufficient to activate Panoptes toxicity. Plasmid identities follow those in 4 d, where T4 Acb2 Y8A is now under the control of the pJEx promoter. (k) N4 Acb4 expression is sufficient to activate Panoptes toxicity. Data from Fig. 4d–f are repeated here for comparative purposes.

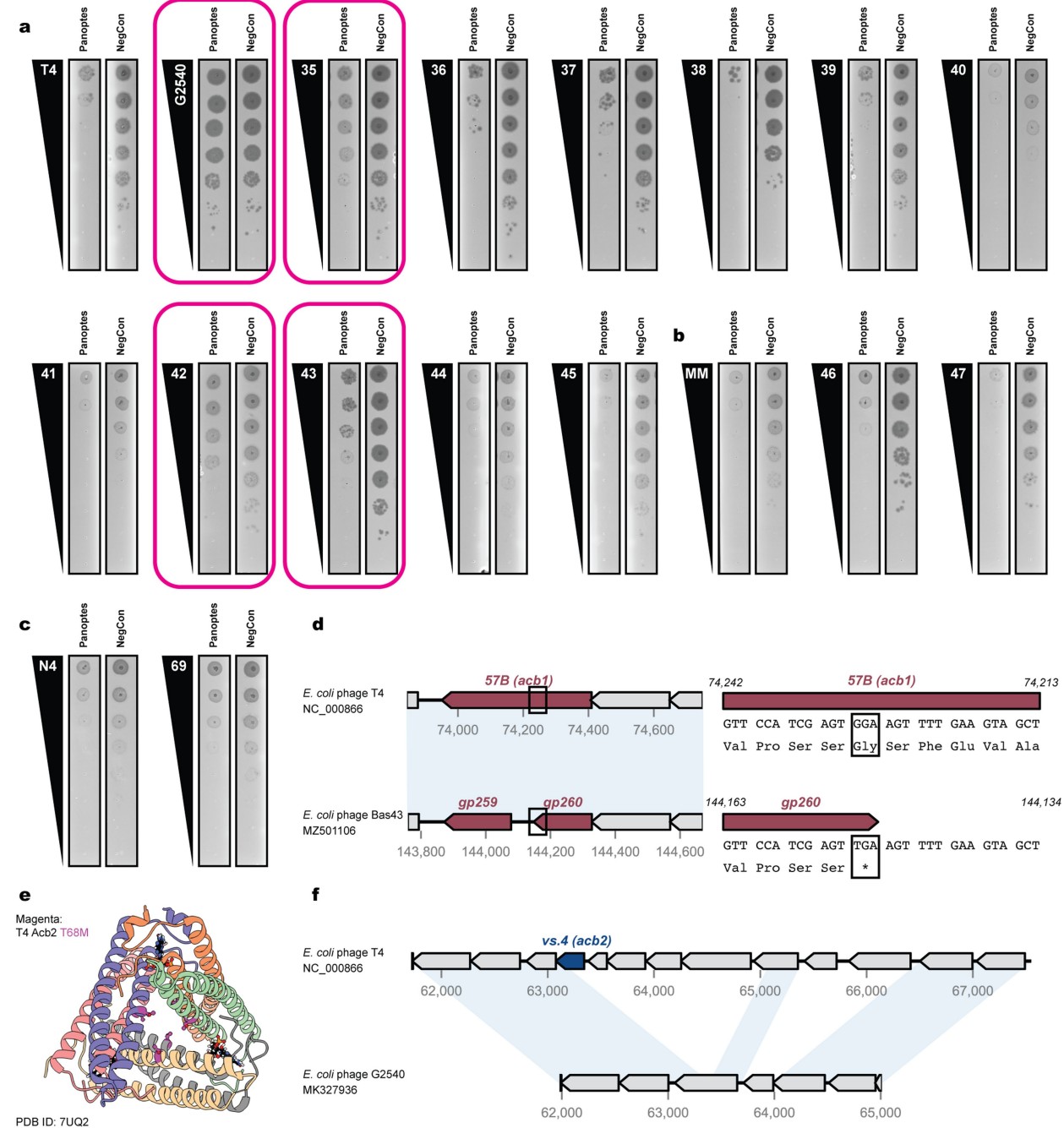

**Extended Data Fig. 7 | Genetic contributions towards phage resistance of Panoptes.** (a) Plaque assays for T4-like phages from DSMZ (G2540) or the BASEL collection (Bas35-45). Resistant phenotypes are highlighted in pink. (b) Plaque assays for MM02-like phages from the BASEL collection (Bas46-47). (c) Plaque assays for N4-like phages from the BASEL collection (Bas69). For panels a-c, plaque assays are representative of 3 biological replicates. (d) Comparative genomics highlights a nonsense mutation at the Bas43 *gp260*

locus (a T4 Acb1 homolog). Regions of high nucleotide similarity as identified by ProgressiveMauve are highlighted. (e) Mutation decreasing T4 Acb2 binding affinity, T68M, in Bas35 shown in magenta overlaid on PDB 7UQ2. (f) Comparative genomics highlights a multigene deletion at the *acb2* locus in between T4 and G2540, including complete loss of *acb2*. Regions of high nucleotide similarity as identified by ProgressiveMauve[34,78] are highlighted.

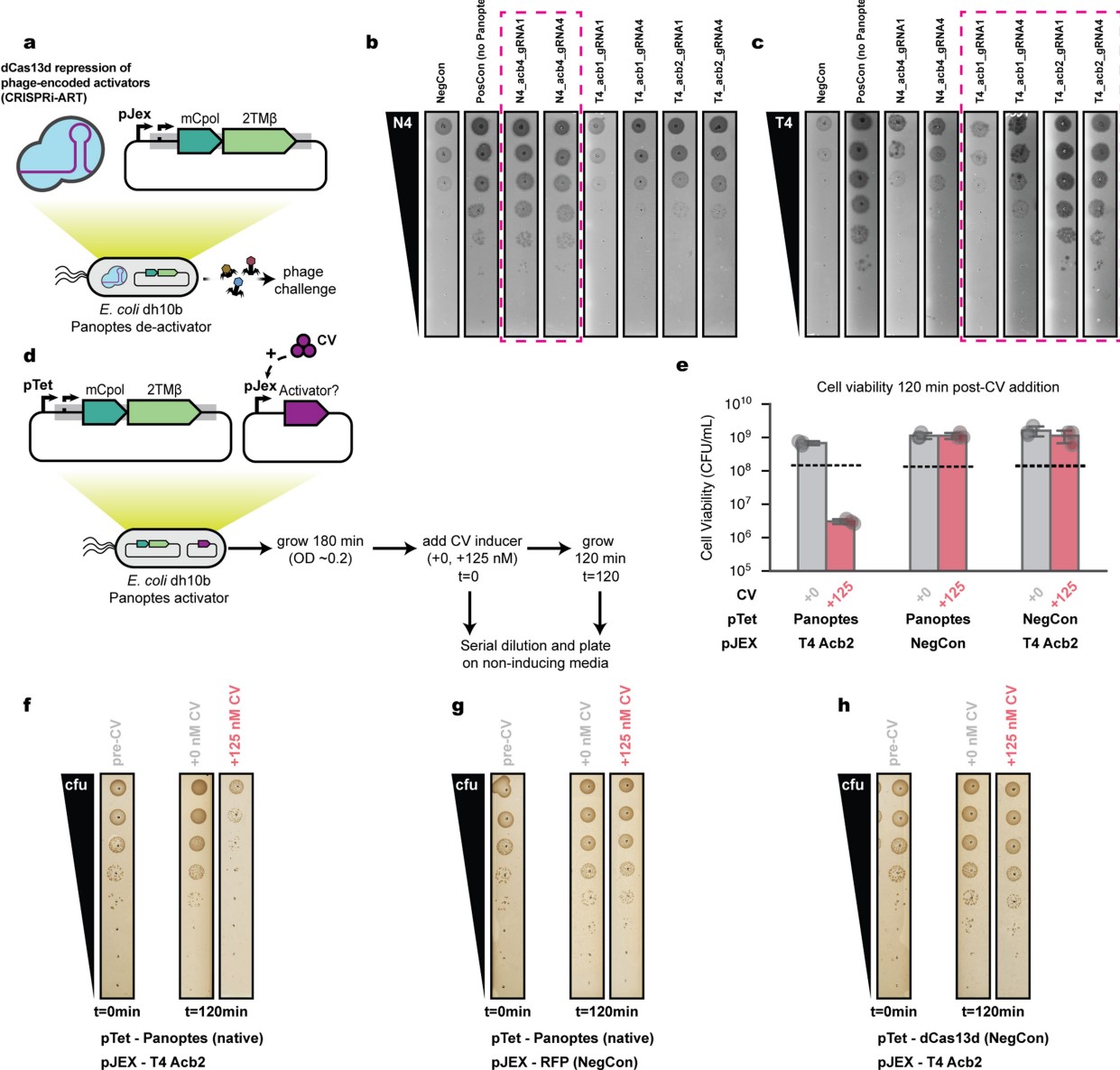

**Extended Data Fig. 8 | Activation of Panoptes by phage encoded anti-CBASS proteins.** (a) CRISPRi-ART screening of phage encoded Acb proteins to restore phage infectivity in the context of Panoptes. CRISPRi-ART uses catalytically deactivated dCas13d to target transcript ribosome binding sites of phage encoded genes to limit their expression[39]. gRNA1 and gRNA4 represent two distinct guide RNAs following the nomenclature from[39]. For all experiments, CRISPRi-ART was expressed at +20 nM aTc. NegCon represents a CRISPRi-ART crRNA targeting a transcript not present in the experiment. PosCon represents a non-targeting CRISPRi-ART crRNA in the absence of Panoptes. CRISPRi-ART conditions targeting a phage-encoded Acb protein in the target phage are bounded in dashed magenta lines. All plaque assays shown are representative images of 3 biological replicates. (b) N4 infection for CRISPRi-ART targeting of Acb proteins in the presence of Panoptes. (c) T4 infection for CRISPRi-ART targeting of Acb proteins in the presence of Panoptes (d) Overview of activator

assay screen to measure bacterial viability. *E. coli* houses two plasmids. One contains either Panoptes with its native promoter or dCas13d (Negative control) cloned under the pTet promoter on a p15a plasmid. The other encodes either T4 Acb2 or RFP (Negative control) under the pJEX promoter on a SC101 plasmid. Cells are grown in LB + antibiotic media for 180 min and CV inducer added at either +0 or +125 nM CV (t = 0). Cells are grown for another 120 min (t = 120). At t = 0 and t = 120 cells are plated on inducer-absent media and grown overnight to measure viability. (e) Estimated viability of activator assay strains after 120 min of growth following +0 nM (gray) or +125 nM (red) CV addition. Dashed lines represent viability at time of CV addition. Samples represent 3 biological replicates from independent overnights and induced cultures. (f-h) Representative viability assays for *E. coli* housing (f) pTet-Panoptes and pJEX-T4 Acb2, (g) pTet-Panoptes and pJEX-RFP (Negative control) and (h) dCas13d (Negative control) and pJEX-T4 Acb2.

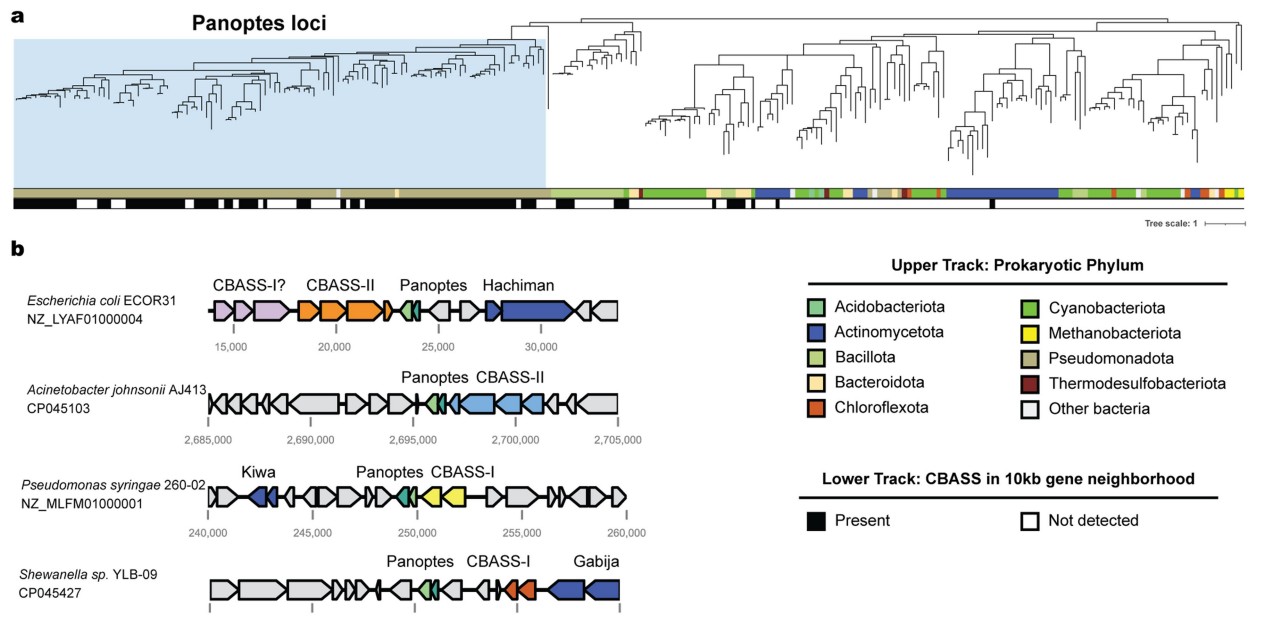

**Extended Data Fig. 9 | Phylogenetic analysis of mCpol proteins and representative Panoptes loci.** (a) Structure-guided phylogenetic tree for the mCpol protein. Top track represents host phylum. Bottom track represents CBASS association in a 10 kb neighborhood of the mCpol-containing protein.

Panoptes clade investigated in this study is highlighted in light blue. (b) Representative Panoptes loci highlighting distinct CBASS systems and other defense systems in the 10 kb gene neighborhood.

**Extended Data Table 1 | X-ray data collection and refinement statistics**

| PDB ID | | mCpol-ApNHpp<br>9NWN |
|---|---|---|
| **Data collection[a,b]** | | |
| Space group | | $P2_12_12$ |
| Cell Dimensions | | |
| | $a, b, c$ (Å) | 49.1, 56.8, 93.7 |
| | $\alpha, \beta, \gamma$ (°) | 90, 90, 90 |
| Resolution (Å) | | 48.55-2.28 (2.37-2.28) |
| $R_{merge}$ (%) | | 14.6 (75.9) |
| $R_{pim}$ (%) | | 6.1 (31.1) |
| $CC_{1/2}$ (%) | | 99.8 (88.4) |
| $<I/\sigma I>$ | | 6.8(2.0) |
| Completeness (%) | | 98.8 (100.0) |
| Redundancy | | 6.6 (6.9) |
| Wilson $B$-factor (Å$^2$) | | 35.28 |
| **Refinement and Validation** | | |
| Resolution (Å) | | 48.55-2.28 (2.37-2.28) |
| Unique Reflections | | 12,282 (1,349) |
| Number of atoms | | |
| | Protein | 1,897 |
| | Ligand | 64 |
| $R_{work}/R_{free}$ (%) | | 22.4/24.9 |
| R.m.s. deviations | | |
| | Bond lengths (Å) | 0.014 |
| | Bond angles (°) | 1.46 |
| Poor rotamers (%) | | 2.35 |
| Ramachandran plot | | |
| | Favored (%) | 94.5 |
| | Allowed (%) | 5.51 |
| | Disallowed (%) | 0 |
| Average $B$-factor (Å$^2$) | | 48.86 |

The values reported correspond to the structure of mCpol bound to ApNHpp (PDB ID: 9NWN). [a]For each structure reported, data were derived from a single crystal. [b]Numbers in parentheses correspond to the highest resolution shell.

# Reporting Summary

## Statistics

For all statistical analyses, confirm that the following items are present in the figure legend, table legend, main text, or Methods section.

| n/a | Confirmed | |
|---|---|---|
| ☐ | ☒ | The exact sample size (*n*) for each experimental group/condition, given as a discrete number and unit of measurement |
| ☐ | ☒ | A statement on whether measurements were taken from distinct samples or whether the same sample was measured repeatedly |
| ☒ | ☐ | The statistical test(s) used AND whether they are one- or two-sided<br>*Only common tests should be described solely by name; describe more complex techniques in the Methods section.* |
| ☒ | ☐ | A description of all covariates tested |
| ☒ | ☐ | A description of any assumptions or corrections, such as tests of normality and adjustment for multiple comparisons |
| ☐ | ☒ | A full description of the statistical parameters including central tendency (e.g. means) or other basic estimates (e.g. regression coefficient) AND variation (e.g. standard deviation) or associated estimates of uncertainty (e.g. confidence intervals) |
| ☒ | ☐ | For null hypothesis testing, the test statistic (e.g. *F*, *t*, *r*) with confidence intervals, effect sizes, degrees of freedom and *P* value noted<br>*Give P values as exact values whenever suitable.* |
| ☒ | ☐ | For Bayesian analysis, information on the choice of priors and Markov chain Monte Carlo settings |
| ☒ | ☐ | For hierarchical and complex designs, identification of the appropriate level for tests and full reporting of outcomes |
| ☒ | ☐ | Estimates of effect sizes (e.g. Cohen's *d*, Pearson's *r*), indicating how they were calculated |

*Our web collection on statistics for biologists contains articles on many of the points above.*

## Software and code

Policy information about availability of computer code

| | |
|---|---|
| Data collection | Mass spectrometry data acquisition was controlled using Xcalibur software version 2.0.7. X-ray data collection was controlled using the BLU-ICE graphical Interface. Size Exclusion Chromatography data acquisition was controlled using the UNICORN 7.3 software from Cytiva. Fluorescence microscoscopy images were collected using using DeltaVision SoftWoRx version 6.5.2. |
| Data analysis | The following software was used: GraphPad Prism 10.2.2, MAFFT v7.490 in Geneious Prime v2023.0.1., Python 3.7.3., Adobe Photoshop (21.2.0) and Adobe Illustrator (24.2). Growth curve and CRISPRi-ART visualizations were performed using the Seaborn v 0.13.2and matplotlib v3.7.2 package in Python. Crystallographic data were processed with the SSRL autoxds script with an I/sigI cutoff ≥ 1.50. Molecular replacement and structural refinement were carried out in PHENIX version 1.21.2-5419. The model was built and adjusted using COOT version 0.9.8.95 EL. Structures were visualized in ChimeraX-1.7.1. Mass spectrometry raw data were converted to mzXML format using msconvert 3.0.19052.1 from the Galaxy platform and data were processed using the open source software MZmine 3.9.0. DefenseFinder Web Service v2.0.0 was used for defense association analysis. Custom Python scripts used for sequence mining, genomic neighborhood analysis, and taxonomic analysis will be available in Zenodo upon publication. |

For manuscripts utilizing custom algorithms or software that are central to the research but not yet described in published literature, software must be made available to editors and reviewers. We strongly encourage code deposition in a community repository (e.g. GitHub). See the Nature Portfolio guidelines for submitting code & software for further information.

## Data

Policy information about **availability of data**

All manuscripts must include a **data availability statement**. This statement should provide the following information, where applicable:
- Accession codes, unique identifiers, or web links for publicly available datasets
- A description of any restrictions on data availability
- For clinical datasets or third party data, please ensure that the statement adheres to our **policy**

> The structure of mCpol-ApNHpp has been deposited under PDB ID: 9NWN and is publicly available. Structures accessed in this work are publicly available under PDB ID: 6IFK, 6IFL and 3W2W. Mass spectrometry data has been deposited at Figshare project number: 261113. Fluorescence microscopy images have been deposited at Figshare (project number: 260219). Plasmids and plasmid sequences are available on Addgene under the article title (Deposition #86474).

## Research involving human participants, their data, or biological material

Policy information about studies with **human participants or human data**. See also policy information about **sex, gender (identity/presentation), and sexual orientation** and **race, ethnicity and racism**.

| | |
|---|---|
| Reporting on sex and gender | N/A |
| Reporting on race, ethnicity, or other socially relevant groupings | N/A |
| Population characteristics | N/A |
| Recruitment | N/A |
| Ethics oversight | N/A |

Note that full information on the approval of the study protocol must also be provided in the manuscript.

# Field-specific reporting

Please select the one below that is the best fit for your research. If you are not sure, read the appropriate sections before making your selection.

☒ Life sciences  ☐ Behavioural & social sciences  ☐ Ecological, evolutionary & environmental sciences

For a reference copy of the document with all sections, see nature.com/documents/nr-reporting-summary-flat.pdf

# Life sciences study design

All studies must disclose on these points even when the disclosure is negative.

| | |
|---|---|
| Sample size | No statistical methods were used to predetermine sample size. Sample sizes were chosen to balance feasibility with information content, while including repetitions to verify reproducibility. We performed experiments in biological triplicate (n = 3) to facilitate measurements of mean and dispersion unless otherwise noted. |
| Data exclusions | No data were excluded from the analyses. |
| Replication | Unless specified otherwise, all reported data represent the average of three biological replicates. All attempts at replication were successful. |
| Randomization | This study did not perform experiments that require sample randomization as this study does not involve allocation into different experimental groups. |
| Blinding | This study did not perform experiments that require blinding since experimental outcomes are objective, quantitative readouts obtained directly from instruments (e.g. growth curves) or using predefined parameters (e.g. plaque quantification). |

# Reporting for specific materials, systems and methods

We require information from authors about some types of materials, experimental systems and methods used in many studies. Here, indicate whether each material, system or method listed is relevant to your study. If you are not sure if a list item applies to your research, read the appropriate section before selecting a response.

## Materials & experimental systems

| n/a | Involved in the study |
|-----|----------------------|
| ☒ ☐ | Antibodies |
| ☒ ☐ | Eukaryotic cell lines |
| ☒ ☐ | Palaeontology and archaeology |
| ☒ ☐ | Animals and other organisms |
| ☒ ☐ | Clinical data |
| ☒ ☐ | Dual use research of concern |
| ☒ ☐ | Plants |

## Methods

| n/a | Involved in the study |
|-----|----------------------|
| ☒ ☐ | ChIP-seq |
| ☒ ☐ | Flow cytometry |
| ☒ ☐ | MRI-based neuroimaging |

## Plants

| | |
|---|---|
| Seed stocks | *Report on the source of all seed stocks or other plant material used. If applicable, state the seed stock centre and catalogue number. If plant specimens were collected from the field, describe the collection location, date and sampling procedures.* |
| Novel plant genotypes | *Describe the methods by which all novel plant genotypes were produced. This includes those generated by transgenic approaches, gene editing, chemical/radiation-based mutagenesis and hybridization. For transgenic lines, describe the transformation method, the number of independent lines analyzed and the generation upon which experiments were performed. For gene-edited lines, describe the editor used, the endogenous sequence targeted for editing, the targeting guide RNA sequence (if applicable) and how the editor was applied.* |
| Authentication | *Describe any authentication procedures for each seed stock used or novel genotype generated. Describe any experiments used to assess the effect of a mutation and, where applicable, how potential secondary effects (e.g. second site T-DNA insertions, mosiacism, off-target gene editing) were examined.* |

