## [Peer Review File · Nature]

A miniature CRISPR-Cas10 enzyme confers immunity by inverse signaling

Corresponding Author: Professor Jennifer Doudna

Version 0:

Reviewer comments:

Referee #1

(Remarks to the Author)

Doherty et al. explore the antiphage defence system mCpol, predicted as ancestral to the type III CRISPR system. Here, the defence system is named MISS and shown to provide protection against a subset of phage including T4. mCpol is shown to synthesize c-di-Amp in vitro (see point 1) and the structure shows a dimeric organisation with nucleotide analogs poised for reaction (Figure 2). Expression of the effector alone in cells is toxic while co-expression with mCpol relieved this effect, akin to a toxin-antitoxin system (figure 3). Sponge proteins and phosphodiesterases were tested and shown to activate MISS, which is proposed as the first example of type IX TA systems where the antitoxin is a small molecule – an intriguing suggestion.

Overall, this is an exciting, novel and unexpected finding, highlighting the complexity of the bacterial immune system.

This reviewer notes a preprint that has explored the same antiphage defence system, and they are largely complimentary and agree on the most important points. There are some differences in the findings that would be valuable to address.

1. If and when these papers are published, I would urge the authors to agree on a single nomenclature? The field would be grateful if this was agreed before publication. This reader prefers Panoptes, for what it's worth.
2. Is the signalling molecule really different in the two studies? It would be important to clarify the exact nature of the signalling nucleotide.
3. The papers differ on the effect of Acb1 on activation of the mCpol system. This could be due to the use of different isomers of c-di-AMP by the two enzymes of course, but this should be investigated.

Main points:

1. The reaction product of mCpol is confirmed as c-di-AMP and 3',3' c-di-AMP is suggested as the relevant isomer. However, two different isomers are possible. To determine which is relevant, run HPLC with the two standards, which have different retention times. The work of Sullivan et al. suggests 2',3' c-di-AMP is at least possible as the relevant product, and perhaps more likely, particularly given that 3',3'-c-di-AMP is used for non-defence pathways in many cell lineages.
2. A "concurrent" cyclase mechanism is suggested (Line 117). Presumably the term "coordinated" is meant, rather than a sequential formation of first one phosphodiester bond and then the other. It seems premature to make this prediction in the absence of any data as it seems plausible that linear PPP-A-p-A molecules could be an intermediate of the reaction, albeit they may remain bound to the mCpol protein, allowing cyclisation.
3. Following on from point 1, if 2',3'-c-di-AMP is confirmed as the signalling molecule, the experiments described in lines 155-163 will need to be repeated / reassessed.

Specific points:

1. An Abi mechanism is proposed, based on high cell death at high MOIs (Figure 1e). However, in the absence of data showing that the kinetics of cell death are accelerated when MISS is present compared to its absence, an Abi mechanism is

not proven.

2. The relationship of mCpol with Cas10 has long been understood – please rephrase the sentence at line 83 accordingly.
3. Line 137. The MISS operon doesn't really act, the MISS system does.
4. Line 160, figure 3f – the experimental procedure used here to observe protein multimers needs at least a quick description in the text or legend.

Referee #2

(Remarks to the Author)

In this paper by Doherty et al., they characterise a new antiphage system. This system, termed MISS, involves a minimal Cas10 palm domain protein (mCpol) that synthesises cyclic-di-AMP. The c-di-AMP inhibits the activity of the protein (2TMbeta) encoded by the second gene in the operon. This protein is a transmembrane protein that they demonstrate is toxic to the cell when it is not inhibited by c-di-AMP produced by mCpol (akin to a toxin-antitoxin system). Toxicity appears to be via disruption of membrane integrity. They show that the system provides defence against phages, including well known phages such as T4 and N4. The system appears to act as an abortive infection system as defence is only provided with MOIs <1. By knocking down or overexpressing different anti-CBASS (Acb) genes in T4 and N4, they demonstrated that multiple Acb proteins are required to trigger the antiphage system. Therefore a model is proposed where the MISS system has evolved to enable phage defence against phages that have evaded CBASS but harboring Acb proteins. Indeed, bioinformatic analyses showed that MISS and CBASS co-occur more than expected in bacterial genomes and also are co-located. The work is interesting, well performed and the conclusions are in line with the results presented.

Major Comments

1. Nature recently released an accelerated article preview on the Hailong defence system, which has a similar de-repression mechanism of activation. The author should include discussion and reference to this system that has a lot of conceptual similarities to the work reported in the current manuscript.
2. I found some of the language and writing in the title, abstract and introduction to be a bit unscientific/journalistic. The results are sound and do not need to be "pumped up" so much. A few examples
 - a. L58 "foils" anti-immunity
 - b. L60 "maintains cell health", whereas it just is locked in an inactive state – not maintaining cell health, just having no negative effect on it, which is true of TA systems in general
 - c. L65 "signalling trap" is confusing as it sounds like it is trapping signals whereas it is more like a sensor for anti-signalling systems.
 - d. The point with the above is that if you just describe your results directly and clearly then it would be less ambiguous to the reader.
3. I do not find the title or the use of inverse signalling to be a useful descriptor of the results of the system discovered. Signalling occurs, is just results in inhibition. The mechanism is disruption of an inhibitory signalling cascade. These are found in all sort of regulatory hierarchies. Suggest rephrasing the title to be more descriptive.
4. Parallels with TA systems in phage defence are drawn, yet none of the primary papers demonstrating toxin-antitoxins provide phage defence are cited (e.g. at L138).
5. The discussion lacks inclusion of the guard hypothesis and other examples that are well known in the literature such as PARIS, Hailong and PrrC.
6. There is no direct evidence presented that the 2TMbeta protein is membrane localised or that it directly responds to the c-di-AMP molecule. Given that these are key parts of the mechanism, stronger evidence in support of these steps would significantly strengthen the manuscript.
7. While the change in oligomeric state is a key aspect of the proposed model, the current evidence presented to support this change could be strengthened (Figure 3F). For example, does a titration of PDE or a time-course show a shift in the multimer band.
8. The model suggests that cell death is occurring due to either cell lysis or membrane depolarisation due to membrane damage. The authors could induce 2TMbeta and see if the OD of a cell culture decreases (lysis) or measure PMF or other membrane permeability metrics.
9. No direct evidence of cell death is presented. Throughout the manuscript, the authors conflate observed changes in OD600 and colony forming units with cell death. CFU cannot distinguish between inhibition of growth (e.g. dormancy or bacteriostasis) or cell killing. There are many standard methods used (suggest the author look at the toxin-antitoxin literature for examples – eg. cell viability stains).
10. The Supplementary File 1 could not be found. In the absence of this file, details of the plasmids used in the study and the methods detailed are insufficient to fully interpret the result or to repeat the experiments.

Minor comments

1. The name of the system "MISS" is not very useful as it not a very unique search term. Also, see above comment about inverse signalling.
2. L42 – remove "expanding our understanding of oligonucleotides in cell health and disease" as this would be misleading to generalist readers and doesn't appear to relate very closely with the work presented.
3. L43-44. This is highly speculative and not relevant for the abstract.
4. L46 – The introduction is light and could be strengthened with further acknowledgement of the existing related literature.
5. L81 – Authors state the results are consistent with the effector being a 2TMB homolog. However, there is no context to the role of 2TMB – see note about a more in depth introduction.
6. L104 – "maintains cell health" see earlier point – please rephrase
7. L112 – The authors state mCpol has conserved Serine residues that are also found in Cas10. Supplementary Figure 5B shows mCpol has S30 and N34. Is N34 conserved?
8. L150 – Authors should state the architecture is from an AlphaFold3 prediction.
9. L188 – The authors do not include the Acb1 result. There appears at least partial protection with Acb1_4.
10. L192, 194 – The authors state "low levels of expression". Compared to what? Were levels measured? Or perhaps a titration of inducer?
11. Figure legend – Many of the legends provide results and insufficient details about the figure panel. All legends should be reviewed and corrected.
12. Figure 2f – Residues should be identified.
13. Figure 3f – is the lower gel a native PAGE? Not explained in legend. Uncropped gels should be made available.
14. Figure 4d-f and Supplementary Figure 10 – the words in the key boxes are difficult to read. Legend requires some description of what this means i.e. what is con and what is the difference between – and +.
15. Figure 4d legend – was this with Acb1_4? Fig 4d-f is an example of the legend containing results and insufficient information detailing the experiment.
16. Figure 4h – it would be good to describe in the main text a little more about what the MISS locus definition was (must these have the 2TMB β effector or are other effectors allowed – e.g. based on mCpol?)
17. Supplementary Figure 1 – define NegCon.
18. Supplementary Figure 2 – The figure is a sequence logo from a sequence alignment. Define "Catalytic residues" and "P-loop residues" in the legend. Legend for b requires further detail.
19. Supplementary Figure 4 – What are the standard sizes?
20. Supplementary Figure 5a – Is N34 also conserved? The authors should extend the sequence logo to show this position.
21. The concept that proteins can respond oppositely (activated or repressed) by small molecular inducers is a well-known concept in regulators of gene expression in bacteria and it might be beneficial to draw that parallel here in the discussion – you can either activate killing or de-repress to result in killing!. Same overall outcome.

Referee #3

(Remarks to the Author)

The authors present a compelling study on the CRISPR-associated enzyme mCpol, which constitutively synthesizes cyclic-di-AMP as part of a novel phage defense system involving the transmembrane effector protein 2TMB β . In this system, cyclic-di-AMP prevents the dimerization of 2TMB β , thereby maintaining cell viability. Upon phage infection, degradation or sequestration of cyclic-di-AMP by phage proteins permits 2TMB β dimerization, ultimately triggering host cell death. The authors support this model with a combination of biochemical, structural, and cellular data. Overall, the manuscript is well written, the experimental evidence is convincing, and the work should be of broad interest. The manuscript is a strong candidate for publication in Nature. Some questions about the biochemical experiments should be addressed before publication.

Comments:

Main text

- Page 5, line 119: "The substrate orientation explains the singular product identity that we observed biochemically." This is problematic as other reaction products are observed in Figure 2A. In the methods section, it says this reaction was incubated for 18 hours. Maybe a shorter incubation would lead to more specific products and the nonspecific products only accumulate after extensive incubation?
- Page 8, line 185: CRISPRi-ART. A reference is given, but it would make the paper more readable to a more general audience if a 1 sentence description of this technology were provided here.
- Page 9, line 207: A 1 sentence description of DefenseFinder should be added to make the paper more readable to a more general audience.
- Figure 1A, right: This panel is somewhat confusing. Under "Cyclase Activation" it would be unclear what the different color blobs are for general readers, especially for the "crRNA binding" blob (that I love). Either more labels should be added to the figure or a more complete description of what each item is in this panel should be added to the figure legend for general readers.
- Related to Figure 2:
 - o There is a control for ATP alone but no control for GTP alone.
 - o Other major reaction products are AMP, ADP, and cAMP, which the authors do not talk about in the main paper. This begs the question as to what effect these reaction products have in a cell. These products are also seen with the D57 active site mutant, which raises the question as to how these products still form. Is there another active site making these products?
 - o The migration of the cyclic-di-AMP standard in the right-most lane (crisp band) in Figure 2A does not seem to match the migration of the diffuse band labeled cyclic-di-AMP in the left-most lane. Furthermore, mass spec does not rule out other cyclic-di-AMP products that are cyclized at a different position. The authors treat the reaction products with S1 nuclease, however, it is unknown how promiscuous this enzyme is in terms of hydrolysis activity on different cyclic nucleotides. A more convincing experiment would be to analyze the 5'-3' cyclic di-AMP standard (and potentially other cyclic di AMP standards) along with the reaction products of mCpol by HPLC as in Figure 2B.
- Figure 3, panel F, TLC analysis. There appears to be a difference in migration between Neg and dPDE. Is this difference real or is there another explanation?

Methods Section

- Page 23, line 460: "E. coli strain ECOR47 was used." A reference is given in the main text for ECOR31, but no reference, source, or genotype is given for ECOR47. More information is needed here.
- Page 23, line 463: "E. coli BW25113". While the genotype is listed, no source or reference is listed here.
- Page 23, line 474: "through electroporation". A reference for this protocol should be listed since it is not a common E coli strain
- Page 23, line 475: "BL21 AI genotype". Please list the source of this strain.
- Page 26, line 576: "pigeon pox virus PDEs were cloned". Please list the source of this gene / sequence.
- Page 26, line 586: "RPM". Please list the value in xg which is standardized across rotors and will increase the reproducibility of the work.
- Page 26, line 602 "using a MWCO centrifugal concentrator". What MWCO concentrator was used? I think this is missing a number in front of MWCO.
- Page 26, line 604 "1 mM TCEP) glycerol monitored". I think the glycerol should be added next to the 10% a few words back.
- Page 26, line 604 "peaks were pooled". Which peaks? The blue trace in Supplementary Figure 4 shows multiple peaks.
- Page 26, line 604 " , concentrated," What concentration was the protein concentrated to? This is crucial for reproducibility since proteins have different solubility limits.
- Page 26, line 611 " , 1g R-250". This should be a percentage or concentration
- Page 27, line 614 " a hydrolysis resistant amine modified analog of c-di-AMP." Is it an analog of c-di-AMP or ATP?
- Page 27, Thin Layer Chromatography (TLC) of cyclase products section:
 - o How was protein concentration determined?
 - o Is it known that vortexing the enzyme stopped the reaction? Some proteins can survive this.
 - o 18 hours seems like an incredibly long incubation time. Is this the source of the other products you see in the TLC

reaction?

o "The reaction mixture was incubated at 37C from 10 min to 18 h and stopped by vortexing" Where is this time course?

• Page 29, Gel shift assay section

o 0.3 mg/ml HIS-GFP-SUMO-2TM β – It would be optimal to list this in micromolar to understand relative concentrations without the need for calculation.

Supplementary Information

• SI Figure 2 – Please add active site residue numbers for the mCpol described in this work either to the figure legend or the main figure so that the readers don't have to search themselves to find this motif in the enzyme. For example, D57 is referenced, but it's difficult to see which residue in either alignment is D57.

• SI Figure 3 – Why does the D57A mutant still make cyclic AMP, AMP, and ADP? Are these promiscuous side reactions or are they important in the phage defense mechanism?

• SI Figure 4

o Panel A – What is the very large peak in blue around 50 ml? Where is the void volume? This is crucial for the interpretation of these results

o Panel C – What is SEC peak 1? There seems to be a small amount of protein that is distributed over multiple bands.

David Taylor and Tyler Dangerfield

Referee #4

(Remarks to the Author)

I co-reviewed this manuscript with one of the reviewers who provided the listed reports.

Referee #5

(Remarks to the Author)

I co-reviewed this manuscript with one of the reviewers who provided the listed reports.

Version 1:

Reviewer comments:

Referee #1

(Remarks to the Author)

The authors have dealt constructively with the reviewers' comments, added further data and clarified points of uncertainty. The correction of the erroneous identification of the c-di-AMP isomer is welcome, as is a common nomenclature for the system. The work is placed more properly in context with prior art.

Specific points:

1. The c-diAMP bound form of the effector is proposed to be a dimer. This is based on elution profile from SEC (Figure 3 and Supplementary figure 10). It is well understood that SEC does not allow accurate determination of molecular weight, and therefore quaternary structure. It seems plausible that the effector is a monomer when inactivated, and indeed this is shown in figure 3g. It is unclear what the structure of this dimer could be, whereas a monomer / multimer transition seems plausible. If the authors insist on this prediction of a dimer, they should provide further diagnostic data to support it. For example, native gel electrophoresis is provided to investigate the large size of the apo-form of the effector (Supplementary figure 10) – this could be extended to the inactivated form. An alternative would be SEC-MALLS. Alternatively, remove the prediction of a dimeric state – it is not important for the overall paper.

2. Line 262-2. The phrase "molecular tripwire" is used twice here in quick succession. Consider rephrasing – once is enough.

3. In the discussion, the power of a reverse-logic signalling "guard" system to shape viral anti-defence is highlighted. The fact remains though that Panoptes is a very rarely-occurring system – the vast majority of CBASS defence systems are not found with Panoptes also present in the genome. This suggests a cost of Panoptes, perhaps due to toxicity, which balances its benefits to the cell. Consider adding a sentence to the discussion to reflect this.

Malcolm White

Referee #3

(Remarks to the Author)

The authors have addressed all of my minor concerns. This is solid and exciting work. I strongly support publication in Nature.

David Taylor and Tyler Dangerfield

Referee #4

(Remarks to the Author)

Overall, Doherty et al. have addressed most of our comments, including textual changes and new data. However, reviewing the textual changes was difficult because the authors did not state the changes made or provide a tracked version of the manuscript.

From their revision, we have the following remarks (point number refers to our original revision):

Major 10) The supplementary file requires replacement of MISS with Panoptes, source of all genes and guide sequences. Although plasmid construction details are absent, the authors state plasmids will be made available through Addgene, from which the plasmid maps should provide some important details.

Minor 4) Our original comment that the introduction is very short and minimizes the contribution of previous work that led to these discoveries has not been adequately addressed. The new discussion relating to other work is appreciated and an improvement.

Minor 12) Residue labels for Figure 2f have not been added.

Minor 13) Replacement of the native gel with the size-exclusion data is an improvement and nicely shows the conversion of the higher oligomeric species to dimer on addition of the signal molecule. Should line 169 be "Supplementary Fig 10c"? In Supp Fig 10d, was this from the Oligomer peak after SEC purification? Otherwise, why isn't the dominant dimer band present? Clarification in the legend would suffice.

Minor 15) The second part of this point was not addressed, where often the authors use the legend to state results and not provide information detailing the experiment. Again, Fig 4d-f are good examples, where these are growth curves. Other legend descriptions have similar issues (e.g. Fig 3 b-d and i-k).

Minor 18) It would be useful to include the P-loop consensus sequence and comment on this compared the obtained sequence logo.

Referee #5

(Remarks to the Author)

I co-reviewed this manuscript with one of the reviewers who provided the listed reports.

Referee #1 (Remarks to the Author):

Doherty et al. explore the antiphage defence system mCpol, predicted as ancestral to the type III CRISPR system. Here, the defence system is named MISS and shown to provide protection against a subset of phage including T4. mCpol is shown to synthesize c-di-Amp in vitro (see point 1) and the structure shows a dimeric organisation with nucleotide analogs poised for reaction (Figure 2). Expression of the effector alone in cells is toxic while co-expression with mCpol relieved this effect, akin to a toxin-antitoxin system (figure 3). Sponge proteins and phosphodiesterases were tested and shown to activate MISS, which is proposed as the first example of type IX TA systems where the antitoxin is a small molecule – an intriguing suggestion.

Overall, this is an exciting, novel and unexpected finding, highlighting the complexity of the bacterial immune system.

This reviewer notes a preprint that has explored the same antiphage defence system, and they are largely complimentary and agree on the most important points. There are some differences in the findings that would be valuable to address.

1. If and when these papers are published, I would urge the authors to agree on a single nomenclature? The field would be grateful if this was agreed before publication. This reader prefers Panoptes, for what it's worth.

We agree that a unified nomenclature between the two contemporary reports is best. We have adopted the Panoptes name in place of MISS throughout our manuscript to accommodate this.

2. Is the signalling molecule really different in the two studies? It would be important to clarify the exact nature of the signalling nucleotide.

We conducted further experiments to verify the identity of the signaling molecule. Although the product of the S1 digestion co-migrated with 5'-AMP by TLC, HPLC-MS of the S1 digestion revealed a molecular weight consistent with pApA: a linear product from incomplete digestion by the 5'-3' specific nuclease. Therefore, we used HPLC-MS of the mCpol-ATP reaction compared with chemical standards to show that the product of mCpol in the presence of ATP is primarily 2'3'-c-di-AMP, with a minor 3'3'-c-di-AMP product. Indeed, we used a thermal shift assay (Supplementary Fig. 10) to determine that 2'3'-c-di-AMP is preferentially bound by the 2TM β effector.

3. The papers differ on the effect of Acb1 on activation of the mCpol system. This could be due to the use of different isomers of c-di-AMP by the two enzymes of course, but this should be investigated.

Both papers have observed that T4 Acb2 contributes more to activation during phage T4 infection as indicated in our work by the larger restoration of T4 infectivity during T4 Acb2 knockdown as compared to T4 Acb1 (Fig. 4b). However, we have observed that the product of mCpol responsible for effector suppression is degraded by T4 Acb1 *in vitro* (Supplementary Fig. 14) and that T4 Acb1 is sufficient to activate the mCpol system in a cellular context (Fig. 4d). Lack of more robust activation by T4 Acb1 as compared to Acb2 may be due to relative enzyme expression levels. For instance, during T4 infection Acb1 is expressed mid-late during infection, whereas Acb2 is one of the first genes expressed (PMID: 36423111). The relative contribution of Acb1 and Acb2 in other phages may differ.

Main points:

1. The reaction product of mCpol is confirmed as c-di-AMP and 3',3' c-di-AMP is suggested as the relevant isomer. However, two different isomers are possible. To determine which is relevant, run HPLC with the two standards, which have different retention times. The work of Sullivan et al. suggests 2',3' c-di-AMP is at least possible as the relevant product, and perhaps more likely, particularly given that 3',3'-c-di-AMP is used for non-defence pathways in many cell lineages.

Please see the response to point #2 for a discussion of the signaling molecule. We used HPLC with c-di-AMP standards to verify the identity of the signaling molecule.

2. A “concurrent” cyclase mechanism is suggested (Line 117). Presumably the term “coordinated” is meant, rather than a sequential formation of first one phosphodiester bond and then the other. It seems premature to make this prediction in the absence of any data as it seems plausible that linear PPP-A-p-A molecules could be an intermediate of the reaction, albeit they may remain bound to the mCpol protein, allowing cyclisation.

Coordinated is indeed a better description for the data we are presenting. The text has been updated accordingly.

3. Following on from point 1, if 2',3'-c-di-AMP is confirmed as the signalling molecule, the experiments described in lines 155-163 will need to be repeated / reassessed.

Since the major product of the mCpol-ATP reaction was confirmed to be 2'3'-c-di-AMP, we conducted multiple additional experiments with the 2TM β Δ 2TM recombinant protein with 2'3'-c-di-AMP. While the protein purified in the presence of 2'3'-c-di-AMP as a dimer (Supplementary Fig. 10b), dialysis of the protein into a buffer lacking the molecule led to higher order oligomeric species as seen by size exclusion chromatography and native PAGE (Fig. 3f; Supplementary

Fig. 10d,e). Additionally, we detected direct binding of 2'3'-c-di-AMP to 2TM β Δ 2TM by conducting a thermal melting assay, which showed significant stabilization of 2TM β Δ 2TM upon the addition of 2'3'-c-di-AMP (Supplementary Fig. 10c). By analytical size exclusion chromatography, we found that titrating increasing amounts of 2'3'-c-di-AMP to 2TM β Δ 2TM leads to an eventual loss of an oligomeric state and clearly demonstrates the dependence of 2TM β multimerization on the signaling molecule (Fig. 3f).

Specific points:

1. An Abi mechanism is proposed, based on high cell death at high MOIs (Figure 1e). However, in the absence of data showing that the kinetics of cell death are accelerated when MISS is present compared to its absence, an Abi mechanism is not proven.

The reviewer points out that an abi mechanism is not proven. As such, we have conducted additional experiments and edited the text to reflect the data we present.

Observing significant cell death and limited phage production at high MOI, but population-level antiphage production at low MOI is consistent with an abi phenotype. We have edited the text surrounding Figure 1 to indicate that what we observe is consistent with an abi phenotype, rather than proving an abi mechanism (PMID: 37268559). Additionally, we have added Supplementary Fig. 16 which shows a decrease in cell viability when Panoptes is activated that is proportional to induction of Panoptes activating proteins. This corroborates a likely abi mechanism, consistent with characterization of the CBASS system harboring the structurally similar effector protein, Cap15. For more discussion of the similarities between the 2TM β effector protein presented in this work and the Cap15 protein, please see response to Reviewer 2, Major comment #6.

2. The relationship of mCpol with Cas10 has long been understood – please rephrase the sentence at line 83 accordingly.

This sentence has been updated to indicate that the active site residue alignment of mCpol is consistent with the known homology to Cas10.

3. Line 137. The MISS operon doesn't really act, the MISS system does.

This is an important distinction in wording; the text has been updated to the "Panoptes system".

4. Line 160, figure 3f – the experimental procedure used here to observe protein multimers needs at least a quick description in the text or legend.

As noted in the response to Main point #3, we have updated the experiment in Fig. 3f to involve the major signaling molecule. These experiments have been described in detail in the text and legend.

Referee #2 (Remarks to the Author):

In this paper by Doherty et al., they characterise a new antiphage system. This system, termed MISS, involves a minimal Cas10 palm domain protein (mCpol) that synthesises cyclic-di-AMP. The c-di-AMP inhibits the activity of the protein (2TMbeta) encoded by the second gene in the operon. This protein is a transmembrane protein that they demonstrate is toxic to the cell when it is not inhibited by c-di-AMP produced by mCpol (akin to a toxin-antitoxin system). Toxicity appears to be via disruption of membrane integrity. They show that the system provides defence against phages, including well known phages such as T4 and N4. The system appears to act as an abortive infection system as defence is only provided with MOIs <1. By knocking down or overexpressing different anti-CBASS (Acb) genes in T4 and N4, they demonstrated that multiple Acb proteins are required to trigger the antiphage system. Therefore a model is proposed where the MISS system has evolved to enable phage defence against phages that have evaded CBASS but harboring Acb proteins. Indeed, bioinformatic analyses showed that MISS and CBASS co-occur more than expected in bacterial genomes and also are co-located. The work is interesting, well performed and the conclusions are in line with the results presented.

Major Comments

1. Nature recently released an accelerated article preview on the Hailong defence system, which has a similar de-repression mechanism of activation. The author should include discussion and reference to this system that has a lot of conceptual similarities to the work reported in the current manuscript.

The Hailong article published during review describes some conceptual similarities with our study. A discussion of this work, along with other recently described bacterial immunity systems that detect the disruption of a host function, have been added to the Discussion.

2. I found some of the language and writing in the title, abstract and introduction to be a bit unscientific/journalistic. The results are sound and do not need to be “pumped up” so much. A few examples

a. L58 “foils” anti-immunity

“Foils” has been replaced with “detects”.

b. L60 “maintains cell health”, whereas it just is locked in an inactive state – not maintaining cell health, just having no negative effect on it, which is true of TA systems in general

This was changed to “represses a toxic effector”.

c. L65 “signalling trap” is confusing as it sounds like it is trapping signals where as it is more like a sensor for anti-signalling systems.

This was changed to “detect phage anti-immunity enzymes and trigger host cell destruction” instead of referring to it as a trap.

d. The point with the above is that if you just describe your results directly and clearly then it would be less ambiguous to the reader.

We appreciate the reviewer's feedback on how to better communicate our results. Language in the title, abstract, and introduction has been updated to more directly describe our findings.

3. I do not find the title or the use of inverse signalling to be a useful descriptor of the results of the system discovered. Signalling occurs, is just results in inhibition. The mechanism is disruption of an inhibitory signalling cascade. These are found in all sort of regulatory hierarchies. Suggest rephrasing the title to be more descriptive.

We appreciate this suggestion from the reviewer and agree that inhibitory signaling is more descriptive of this mechanism. This change has been reflected in the title and throughout the manuscript.

4. Parallels with TA systems in phage defence are drawn, yet none of the primary papers demonstrating toxin-antitoxins provide phage defence are cited (e.g. at L138).

We have updated the references to include a recent review (in the interest of the main text citation limit) that includes primary papers demonstrating that toxin-antitoxin systems provide phage defense.

5. The discussion lacks inclusion of the guard hypothesis and other examples that are well known in the literature such as PARIS, Hailong and PrrC.

The Discussion has been edited to include a description of other bacterial immune systems that exemplify the guard hypothesis, and parallels with the Hailong system.

6. There is no direct evidence presented that the 2TM β protein is membrane localised or that it directly responds to the c-di-AMP molecule. Given that these are key parts of the mechanism, stronger evidence in support of these steps would significantly strengthen the manuscript.

A related 2TM β effector, Cap15, was shown to induce membrane depolarization upon activation by a CBASS-produced molecule (Duncan-Lowey et al., 2021). While the 2TM β protein presented in this work displays an inverted response to the signaling molecule when compared to Duncan-Lowey et al., it is nonetheless similar by several important metrics. We now highlight these similarities and clarify ambiguity.

The mCpol-associated 2TM β effector studied here possesses a high degree of structural homology to the CBASS-associated Cap15 effector from Duncan-Lowey et al., 2021, with a DALI Z-score of 16, which is well-above the significant homology cutoff of 2 (PMID: 18818215). Notably, the topological arrangement of predicted transmembrane helices are highly conserved as by TMHMM (PMID: 18818215) between these two proteins. This analysis is represented in a new Supplementary Figure 8 added to the revision manuscript. In addition, our microscopy data from the initial submission shows remarkable similarity in the morphology of membrane lesions

upon activation of Cap15 and the 2TM β from this work. Given these findings, membrane association would not be surprising or informative beyond data shown in Duncan-Lowey et al., 2021. Instead we feel that highlighting the differences in regulation is most relevant for this current study.

As direct response of 2TM β to the c-di-AMP molecule is a key part of our findings and a distinguishing feature from Duncan-Lowey et al., 2021, we have included additional experiments illustrating the multimerization state change in response to 2'3'-c-di-AMP by thermal shift assay and analytical size exclusion chromatography. Please see response to Reviewer 1, Main point #3 for a more detailed discussion of these experiments.

7. While the change in oligomeric state is a key aspect of the proposed model, the current evidence presented to support this change could be strengthened (Figure 3F). For example, does a titration of PDE or a time-course show a shift in the multimer band.

We have significantly strengthened the characterization of c-di-AMP control of the 2TM β oligomeric state, including a titration of 2'3'-c-di-AMP by analytical size exclusion chromatography which shows loss of a multimeric peak with increasing amounts of the inhibitor molecule (Fig. 3f). For a more extensive discussion of this new work please see response to Reviewer 1, Main point #3.

8. The model suggests that cell death is occurring due to either cell lysis or membrane depolarisation due to membrane damage. The authors could induce 2TM β and see if the OD of a cell culture decreases (lysis) or measure PMF or other membrane permeability metrics.

The original manuscript included microscopy data obtained using a membrane stain (Fig. 3) that revealed membrane-lesion effects consistent with those observed for the 2TM β protein studied in Duncan-Lowey et al., 2021. We now also include new experiments to test cell death, as described in Reviewer 2, point #9.

9. No direct evidence of cell death is presented. Throughout the manuscript, the authors conflate observed changes in OD600 and colony forming units with cell death. CFU cannot distinguish between inhibition of growth (e.g. dormancy or bacteriostasis) or cell killing. There are many standard methods used (suggest the author look at the toxin-antitoxin literature for examples – eg. cell viability stains).

We measured cell viability following activation of mCpol-2TM β , revealing that activation by T4Acb2 for 2 hours lowers viable colony forming unit counts from the time of induction even after removal from inducing media. This decrease in cell viability was not attributable to T4 Acb2 toxicity (in the absence of mCpol-2TM β), mCpol-2TM β to the inducer (in the absence of T4 Acb2) or other cell death-promoting factors during phage infection. These data demonstrate that the effect of mCpol-2TM β is indeed primarily cell-death versus dormancy.

Additionally, we use cell viability staining (SYTOX Green) to show evidence of 2TM β -induced cell death. Consistent with the membrane lesions that develop when 2TM β is expressed without mCpol (Fig. 3i,j), we observed cell lysis dependent on 2TM β in the absence of mCpol expression (Supplementary Fig. 11). Collectively, these results demonstrate that 2TM β induces cell death.

10. The Supplementary File 1 could not be found. In the absence of this file, details of the plasmids used in the study and the methods detailed are insufficient to fully interpret the result or to repeat the experiments.

Supplementary File 1 was mistakenly not included in the original submission and has been uploaded for review.

Minor comments

1. The name of the system “MISS” is not very useful as it not a very unique search term. Also, see above comment about inverse signalling.

We have revised the manuscript to adopt the Panoptes nomenclature, and have adopted the term “inhibitory signaling” which is a better fit for the mechanism that we describe. Please see response to Reviewer.

2. L42 – remove “expanding our understanding of oligonucleotides in cell health and disease” as this would be misleading to generalist readers and doesn’t appear to relate very closely with the work presented.

We have rephrased this to “expanding our understanding of the role of oligonucleotides in immunity” to more specifically represent how this work expands upon existing knowledge.

3. L43-44. This is highly speculative and not relevant for the abstract.

This sentence has been omitted from the abstract.

4. L46 – The introduction is light and could be strengthened with further acknowledgement of the existing related literature.

Although we have added several citations to the introduction (phage encoded Acb sponge proteins), in the interest of word count, we have added a more in-depth discussion of relevant literature to the Discussion section of the manuscript. Please see response to Major point #5.

5. L81 – Authors state the results are consistent with the effector being a 2TMB homolog. However, there is no context to the role of 2TMB – see note about a more in depth introduction.

We have added a reference to Cap15, the homolog of the 2TM β , and its role in phage infection as determined in previous literature.

6. L104 – “maintains cell health” see earlier point – please rephrase

This phrasing has been changed to indicate that signaling by mCpol inhibits the 2TM β effector.

7. L112 – The authors state mCpol has conserved Serine residues that are also found in Cas10. Supplementary Figure 5B shows mCpol has S30 and N34. Is N34 conserved?

The phrasing has been updated to indicate that S30 is conserved, consistent with the serine at the homologous position in Cas10. We have included explicitly in the text that S30 and N34 make contacts with the adenine base in ECOR31 mCpol. Supplementary Fig. 6 (Previously Supplementary Fig. 5) has been updated to show conservation of the position corresponding to N34. Please see response to Reviewer 2 minor comment #20 for a discussion of N34 conservation.

8. L150 – Authors should state the architecture is from an AlphaFold3 prediction.

This line has been updated to indicate that the architecture is from an AlphaFold3 prediction.

9. L188 – The authors do not include the Acb1 result. There appears at least partial protection with Acb1_4.

A line has been added to indicate that T4 Acb1 partially restored infectivity, consistent with our in vitro data. Please see the response to Reviewer 1, point #3 for a more detailed discussion of T4 Acb1 activation of the Panoptes system.

10. L192, 194 – The authors state “low levels of expression”. Compared to what? Were levels measured? Or perhaps a titration of inducer?

Low levels of expression are indeed based on the amount of inducer used in these experiments. This experiment induced the pJEx promoter at 125 nM crystal violet. The concentration found to fully induce gene expression is ~1 μ M crystal violet (PMID: 30190458). We have added a citation to the report characterizing this promoter.

11. Figure legend – Many of the legends provide results and insufficient details about the figure panel. All legends should be reviewed and corrected.

All figure legends have been reviewed, and additional detail has been added to properly explain the figure panels.

12. Figure 2f – Residues should be identified.

Residue labels have been added to Figure 2f and g.

13. Figure 3f – is the lower gel a native PAGE? Not explained in legend. Uncropped gels should be made available.

The native PAGE from the original submission was omitted in favor of more relevant experiments demonstrating a direct interaction between the 2TM β and the major signaling molecule. However, uncropped gels and TLCs are available in Supplementary Figure 18.

14. Figure 4d-f and Supplementary Figure 10 – the words in the key boxes are difficult to read. Legend requires some description of what this means i.e. what is con and what is the difference between – and +.

Font color was updated for readability; -con was changed to NegCon for consistency with other figures, and detail has been added to the legends. Note that Supplementary Figure 10 is now Supplementary Figure 15.

15. Figure 4d legend – was this with Acb1_4? Fig 4d-f is an example of the legend containing results and insufficient information detailing the experiment.

Figure 4d is related to the Figure 4c activator assays–this clarification has been made in the legend. The legend has been edited to describe the experimental design of CRISPRi-ART and the activator experiments.

16. Figure 4h – it would be good to describe in the main text a little more about what the MISS locus definition was (must these have the 2TMbeta effector or are other effectors allowed – e.g. based on mCpol?)

This information has been added to the figure caption to indicate that the Panoptes system requires both mCpol and 2TM β . Additionally, we added a reference to Supplementary Figure 17 which shows representative loci.

17. Supplementary Figure 1 – define NegCon.

This has been added to the figure legend.

18. Supplementary Figure 2 – The figure is a sequence logo from a sequence alignment. Define “Catalytic residues” and “P-loop residues” in the legend. Legend for b requires further detail.

The figure legend has been corrected to indicate that this is indeed a sequence logo, and we have defined catalytic residues and P-loop residues in the legend. Alignment position and the corresponding residue in ECOR31 mCpol have been added to the legend.

19. Supplementary Figure 4 – What are the standard sizes?

The standard sizes have been added to the figure legend. Please note that this is now Supplementary Figure 5.

20. Supplementary Figure 5a – Is N34 also conserved? The authors should extend the sequence logo to show this position.

N34 is not highly conserved in our alignment. The alignment in Supplementary Figure 6 (previously Supplementary Figure 5) has been extended to include this position and the figure legend has been updated to indicate the placement of S30 and N34 from ECOR31 within the alignment.

21. The concept that proteins can respond oppositely (activated or repressed) by small molecular inducers is a well-known concept in regulators of gene expression in bacteria and it might be beneficial to draw that parallel here in the discussion – you can either activate killing or de-repress to result in killing!. Same overall outcome.

We agree that there are fascinating parallels in the mechanistic flexibility of small-molecule binding proteins. We included a statement in the Discussion highlighting how non-immune cyclic-di-GMP signaling can have activating and inhibiting mechanisms.

Referee #3 (Remarks to the Author):

The authors present a compelling study on the CRISPR-associated enzyme mCpol, which constitutively synthesizes cyclic-di-AMP as part of a novel phage defense system involving the transmembrane effector protein 2TM β . In this system, cyclic-di-AMP prevents the dimerization of 2TM β , thereby maintaining cell viability. Upon phage infection, degradation or sequestration of cyclic-di-AMP by phage proteins permits 2TM β dimerization, ultimately triggering host cell death. The authors support this model with a combination of biochemical, structural, and cellular data. Overall, the manuscript is well written, the experimental evidence is convincing, and the work should be of broad interest. The manuscript is a strong candidate for publication in Nature. Some questions about the biochemical experiments should be addressed before publication.

Comments:

Main text

1. Page 5, line 119: "The substrate orientation explains the singular product identity that we observed biochemically." This is problematic as other reaction products are observed in Figure 2A. In the methods section, it says this reaction was incubated for 18 hours. Maybe a shorter incubation would lead to more specific products and the nonspecific products only accumulate after extensive incubation?

The sentence has been re-written as "This substrate orientation explains the lack of higher order cyclic oligoadenylates which are synthesized by Cas10." to better convey that the difference between mCpol and Cas10 is the lack of oligoadenylates larger than dinucleotides.

Additionally, we have included a time course in the mCpol-ATP reaction to Supplemental Figure 3. This shows that the product distribution is not significantly changed with shorter incubation periods.

2. Page 8, line 185: CRISPRi-ART. A reference is given, but it would make the paper more readable to a more general audience if a 1 sentence description of this technology were provided here.

A sentence describing CRISPRi-ART technology has been added to the text.

3. Page 9, line 207: A 1 sentence description of DefenseFinder should be added to make the paper more readable to a more general audience.

An introduction to DefenseFinder has been added to the text.

4. Figure 1A, right: This panel is somewhat confusing. Under “Cyclase Activation” it would be unclear what the different color blobs are for general readers, especially for the “crRNA binding” blob (that I love). Either more labels should be added to the figure or a more complete description of what each item is in this panel should be added to the figure legend for general readers.

Labels have been added to Figure 1a to indicate the identity of proteins and protein complexes involved in the cyclase activation step.

Related to Figure 2:

5. There is a control for ATP alone but no control for GTP alone.

The control for GTP alone has been added to Figure 2a, alongside the mCpol-GTP condition.

6. Other major reaction products are AMP, ADP, and cAMP, which the authors do not talk about in the main paper. This begs the question as to what effect these reaction products have in a cell. These products are also seen with the D57 active site mutant, which raises the question as to how these products still form. Is there another active site making these products?

mCpol D57A is unable to suppress 2TM β -induced toxicity (Fig. 3d), indicating that these products are not involved in the inhibitory signaling mechanism described in this work. The role, if any, of cyclic AMP, AMP, and ADP remains to be seen.

7. The migration of the cyclic-di-AMP standard in the right-most lane (crisp band) in Figure 2A does not seem to match the migration of the diffuse band labeled cyclic-di-AMP in the left-most lane. Furthermore, mass spec does not rule out other cyclic-di-AMP products that are cyclized at a different position. The authors treat the reaction products with S1 nuclease, however, it is unknown how promiscuous this enzyme is in terms of hydrolysis activity on different cyclic nucleotides. A more convincing experiment would be to analyze the 5'-3' cyclic di-AMP standard (and potentially other cyclic di AMP standards) along with the reaction products of mCpol by HPLC as in Figure 2B.

Indeed, the major product was confirmed to be 2'3'-c-di-AMP by HPLC-MS as suggested. Please see response to Reviewer #1 for a more complete discussion of this point.

8. Figure 3, panel F, TLC analysis. There appears to be a difference in migration between Neg and dPDE. Is this difference real or is there another explanation?

Both digestion samples display the same migration when co-spotted. However, this experiment has been replaced with more relevant work detailed in response to Reviewer 1, Main point #3.

Methods Section

1. Page 23, line 460: "E. coli strain ECOR47 was used." A reference is given in the main text for ECOR31, but no reference, source, or genotype is given for ECOR47. More information is needed here.

ECOR47 is a member of the well-known *E. coli* reference collection, ECOR. We have added a reference to the ECOR collection for ECOR47.

2. Page 23, line 463: "E. coli BW25113". While the genotype is listed, no source or reference is listed here.

The reference for *E. coli* BW25113 has been added to the text.

3. Page 23, line 474: "through electroporation". A reference for this protocol should be listed since it is not a common *E. coli* strain

We have included the source of these competent cells, Intact genomics, and a protocol for electroporation.

4. Page 23, line 475: "BL21 AI genotype". Please list the source of this strain.

The text has been updated with the manufacturer of this strain.

5. Page 26, line 576: "pigeon pox virus PDEs were cloned". Please list the source of this gene / sequence.

Given the revelation that the major product of 2TM β is indeed 2'3'-c-di-AMP, the gel shift assay was replaced by a titration of 2'3'-c-di-AMP to 2TM β (Δ TM). As such, the methods discussed here have been removed.

6. Page 26, line 586: "RPM". Please list the value in xg which is standardized across rotors and will increase the reproducibility of the work.

All instances of rpm have been converted to x g.

7. Page 26, line 602 "using a MWCO centrifugal concentrator". What MWCO concentrator was used? I think this is missing a number in front of MWCO.

Indeed this was missing the molecular weight specification, this has been added to the text.

8. Page 26, line 604 " 1 mM TCEP) glycerol monitored". I think the glycerol should be added next to the 10% a few words back.

This typo has been corrected.

9. Page 26, line 604 “peaks were pooled”. Which peaks? The blue trace in Supplementary Figure 4 shows multiple peaks.

The relevant elution volume in mL has been added to the methods for each purified protein. Please note that this is now Supplementary Figure 5.

10. Page 26, line 604 “, concentrated,” What concentration was the protein concentrated to? This is crucial for reproducibility since proteins have different solubility limits.

Final protein concentration has been added to the text.

11. Page 26, line 611 “, 1g R-250”. This should be a percentage or concentration

We have corrected this line to read 0.1% R-250.

12. Page 27, line 614 “ a hydrolysis resistant amine modified analog of c-di-AMP.” Is it an analog of c-di-AMP or ATP?

We appreciate the reviewer’s attention to detail, as we meant an analog of ATP. This correction has been made in the text.

13. Page 27, Thin Layer Chromatography (TLC) of cyclase products section:
o How was protein concentration determined?

Protein concentration was determined by absorbance at 280 nm. The “Protein expression and purification” section of the text has been updated to include this.

14. Is it known that vortexing the enzyme stopped the reaction? Some proteins can survive this.

The procedure has been conducted where the reaction was stopped by either vortexing or heating the reaction. In both cases, visible precipitation is induced, indicating that the mCpol is denatured.

15. 18 hours seems like an incredibly long incubation time. Is this the source of the other products you see in the TLC reaction?

We have added a time course of the mCpol-ATP reaction to Supplemental Figure 3. This shows that the product distribution is consistent as a function of time. The reaction is allowed to continue overnight to maximize the conversion of ATP to product.

16. “The reaction mixture was incubated at 37C from 10 min to 18 h and stopped by vortexing”

Where is this time course?

The time course was originally omitted and has been added to Supplementary Figure 3.

• Page 29, Gel shift assay section

17. 0.3 mg/ml HIS-GFP-SUMO-2TM β – It would be optimal to list this in micromolar to understand relative concentrations without the need for calculation.

Given verification that the major product of 2TM β is 2'3'-c-di-AMP, the gel shift assay was replaced by a titration of 2'3'-c-di-AMP to 2TM β (Δ TM). As such, the methods discussed here have been removed. However, we found this to be a pertinent suggestion and concentrations of both 2TM β (Δ TM) and the signaling molecule are included in micromolar.

Supplementary Information

1. SI Figure 2 – Please add active site residue numbers for the mCpol described in this work either to the figure legend or the main figure so that the readers don't have to search themselves to find this motif in the enzyme. For example, D57 is referenced, but it's difficult to see which residue in either alignment is D57.

Active site residue numbers for the mCpol described in this work have been added to the figure legend so that it is clear which residues align with those references within the manuscript.

2. SI Figure 3 – Why does the D57A mutant still make cyclic AMP, AMP, and ADP? Are these promiscuous side reactions or are they important in the phage defense mechanism?

Please see our response to Reviewer 3, Main text #6 for a more complete discussion of the signaling molecule that this work indicates is relevant to the phage defense mechanism.

3. SI Figure 4

o Panel A – What is the very large peak in blue around 50 ml? Where is the void volume? This is crucial for the interpretation of these results

The large peak in blue at ~48 mL represents the void volume of the column. This trace is from the purification of mCpol and not from a previously purified sample. This clarification has been added to the figure and figure legend of the SI.

4. Panel C – What is SEC peak 1? There seems to be a small amount of protein that is distributed over multiple bands.

SEC peak 1 consists of the void volume of the column. As it is not relevant for this study, this label has been removed.

David Taylor and Tyler Dangerfield

Referee #4 (Remarks to the Author):

I co-reviewed this manuscript with one of the reviewers who provided the listed reports.

Referee #5 (Remarks to the Author):

I co-reviewed this manuscript with one of the reviewers who provided the listed reports.

This email has been sent through the Springer Nature Manuscript Tracking System NY-610A-SN&MTS

Confidentiality Statement:

We appreciate the reviewers' critiques and the opportunity to revise the manuscript accordingly. We have addressed the remaining points from reviewers and provided a Word document with tracked changes. Our response to the remaining points from reviewers is as follows:

Referee #1 (Remarks to the Author):

The authors have dealt constructively with the reviewers' comments, added further data and clarified points of uncertainty. The correction of the erroneous identification of the c-di-AMP isomer is welcome, as is a common nomenclature for the system. The work is placed more properly in context with prior art.

Specific points:

1. The c-diAMP bound form of the effector is proposed to be a dimer. This is based on elution profile from SEC (Figure 3 and Supplementary figure 10). It is well understood that SEC does not allow accurate determination of molecular weight, and therefore quaternary structure. It seems plausible that the effector is a monomer when inactivated, and indeed this is shown in figure 3g. It is unclear what the structure of this dimer could be, whereas a monomer / multimer transition seems plausible. If the authors insist on this prediction of a dimer, they should provide further diagnostic data to support it. For example, native gel electrophoresis is provided to investigate the large size of the apo-form of the effector (Supplementary figure 10) – this could be extended to the inactivated form. An alternative would be SEC-MALLS. Alternatively, remove the prediction of a dimeric state – it is not important for the overall paper.

We agree with the assessment that classification of the lower order state is not important for the paper's overall conclusions and thus have removed references to the dimer in the text.

2. Line 262-2. The phrase "molecular tripwire" is used twice here in quick succession. Consider rephrasing – once is enough.

The second instance of the phrase "molecular tripwire" has been removed from the text to improve readability.

3. In the discussion, the power of a reverse-logic signalling "guard" system to shape viral anti-defence is highlighted. The fact remains though that Panoptes is a very rarely-occurring system – the vast majority of CBASS defence systems are not found with Panoptes also present in the genome. This suggests a cost of Panoptes, perhaps due to toxicity, which balances its benefits to the cell. Consider adding a sentence to the discussion to reflect this.

We have added a sentence to the discussion to discuss the potential toxicity of Panoptes limiting the frequency of its occurrence.

Malcolm White

Referee #3 (Remarks to the Author):

The authors have addressed all of my minor concerns. This is solid and exciting work. I strongly support publication in Nature.

David Taylor and Tyler Dangerfield

Referee #4 (Remarks to the Author):

Overall, Doherty et al. have addressed most of our comments, including textual changes and new data. However, reviewing the textual changes was difficult because the authors did not state the changes made or provide a tracked version of the manuscript.

From their revision, we have the following remarks (point number refers to our original revision):

Major 10) The supplementary file requires replacement of MISS with Panoptes, source of all genes and guide sequences. Although plasmid construction details are absent, the authors state plasmids will be made available through Addgene, from which the plasmid maps should provide some important details.

All remaining instances of MISS in the Supplementary file have been removed, guide sequences and gene source was specified. The plasmids will be made available through Addgene.

Minor 4) Our original comment that the introduction is very short and minimizes the contribution of previous work that led to these discoveries has not been adequately addressed. The new discussion relating to other work is appreciated and an improvement.

We have updated the introduction to narrate the discovery of oligonucleotide signaling pathways and emphasize the bioinformatic work that led to predictions about the function of mCpol in defense.

Minor 12) Residue labels for Figure 2f have not been added.

Figure 2f was updated to include labels of “ATP” and “ATP analog” which is now also defined in the legend.

Minor 13) Replacement of the native gel with the size-exclusion data is an improvement and nicely shows the conversion of the higher oligomeric species to dimer on addition of the signal molecule. Should line 169 be “Supplementary Fig 10c”? In Supp Fig 10d, was this from the Oligomer peak after SEC purification? Otherwise, why isn't the dominant dimer band present? Clarification in the legend would suffice.

Line 169 was corrected to Supplementary Fig. 10c and we updated the Supplementary Figure 10d legend to indicate that the band represents the 44 kDa peak from SEC.

Minor 15) The second part of this point was not addressed, where often the authors use the legend to state results and not provide information detailing the experiment. Again, Fig 4d-f are good examples, where these are growth curves. Other legend descriptions have similar issues (e.g. Fig 3 b-d and i-k).

Figure 4d-f, 3b-d and i-k have been corrected to explicitly define the figure as opposed to stating results.

Minor 18) It would be useful to include the P-loop consensus sequence and comment on this compared the obtained sequence logo.

The P-loop consensus sequence was added to the figure legend of Supplementary Fig. 2 and a comment was added to note that the last two residues are less strictly conserved.